

# A Tri-Approach for Diagnosing Gridded Precipitation Datasets for Watershed Glacio-Hydrological Simulation in Mountain Regions

Muhammad Shafeeque[1,2], Luo Yi[1,2,3]

[1]Key Lab of Ecosystem Network Observation and Modelling, Institute of Geographic Sciences and Natural Resources
Research, Chinese Academy of Sciences, 100101 Beijing, China
[2]University of Chinese Academy of Sciences, 100049 Beijing, China
[3]CAS Research Center for Ecology and Environment of Central Asia, Urumqi, China

*Correspondence to*: Luo Yi (luoyi@igsnrr.ac.cn)

**Abstract.** In mountain regions, validation and local correction of gridded precipitation datasets (GPDs) are pre-requisites for
glacio-hydrological simulations. However, insufficient observed data and glacial involvement make it a complicated task in
glacierized watersheds. To diagnose the potential problems in GPDs from multiple perspectives and provide directions for
their correction, a Tri-approach framework, consisting of statistical analysis, physical diagnosis, and practical simulation, is
proposed. Truc-Budyko theory is introduced into this framework, which can identify the actual under- or over-estimation of
GPDs based on watershed water-energy balance, diagnose their possible causes, and provide directions for local correction.
This framework was applied to the glacierized Upper Indus Basin (UIB) for evaluating GPDs, including APHRODITE,
CFSR, PGMFD, TRMM, and HAR, against adjusted observed precipitation (OBS), specific runoff, and glacier mass balance
over varying periods during 1951−2017. The Spatial Processes in HYdrology (SPHY) model was used to simulate the
hydrology and glacier changes (2001−2007). The results suggest that (a) patterns of inter- and intra-annual variations of OBS
precipitation were better captured by APHRODITE (CC >0.6), but it was underestimated (-40%), (b) UIB was characterized
as "Leaky" catchment based on overestimated CFSR (106%) and HAR (77%), indicating positive glacier storage changes
(0.37 and 0.21 m w.e. yr[-1], respectively). In contrast, UIB was characterized as "Gaining" watershed for remaining
underestimated datasets, indicating negative storage changes (-0.42 to -0.34 m w.e. yr[-1]). (c) For constant mass balance, the
simulated runoff was overestimated in SPHY_CFSR (66%) and SPHY_HAR (53%), whereas it was underestimated for
SPHY_APHRODITE (-41%), SPHY_PGMFD (-26%), and SPHY_TRMM (-33%). It highlights that evaluated GPDs could
not generally meet the requirements of the rational output of glacier mass balance and streamflow concurrently. The physical
diagnosis directs local correction based on under- and over-estimation. The practical simulation explores the extent of
expected uncertainties in intra/inter-annual characteristics of glacio-hydrology.

## 1 Introduction

Hydrological simulation is almost inevitably used to develop policies and pro-active remediation and mitigation of the
challenges caused by climate change and its impacts on river basins nowadays (Immerzeel et al., 2020; Pritchard, 2019). The





uncertainties in the hydrological simulation results may affect the understanding and management strategies of the water resources, and so, the millions of people who are dependent on these resources for hydropower generation, domestic, industries, agriculture needs, and maintaining the ecological systems (Huss and Hock, 2018; Luo et al., 2018; Singh et al., 2016; Viviroli et al., 2007). Precipitation is the main input to the hydrological models, and the availability of accurate

precipitation data is the pre-requisite of efficient simulation and reliable results in any catchment (Koutsouris et al., 2016; Tong et al., 2014). Meanwhile, it is a fact that observed precipitation data are usually insufficient or even unavailable for effective hydrological simulation, especially in the high elevation zones and where glaciers exist. Fortunately, to overcome the problem of data scarcity and distribution biasness, scientists have developed gridded precipitation datasets (GPDs) using observed, reanalysis, or remote sensing data or a combination of them (Dee et al., 2011; Harris et al., 2014; Huffman and

Bolvin, 2015; Maussion et al., 2014; Saha et al., 2010; Yatagai et al., 2012). These datasets provide continuous coverage of land, usually span from the 1960s up today with daily time step, and thus provide an opportunity for driving the hydrological simulation in the data scare watersheds.

The accuracy of GPDs is subjected to certain uncertainties due to their methods of development, sources, spatial, and temporal resolution coverage (Sun et al., 2018). Each kind of dataset has its own advantages and limitations. For example,

relatively high accuracy in rain gauge observations, whereas, at the same time, uneven and insufficient distribution of meteorological networks is a limitation in such datasets (Ji et al., 2020; Salio et al., 2015; Woldemeskel et al., 2013). The reanalysis output-based datasets have advantageous in their high spatial and temporal resolutions, but they have limitations linked to a high level of uncertainties due to the issues in input data and methods adopted in reanalysis (Knutti and Sedláček, 2012). Similarly, satellite-based datasets are advantageous in their resolution, but they are limited due to uncertainties caused

by the sensors and algorithms used to retrieve and estimate precipitation, respectively (Sorooshian et al., 2011). Regional studies showed the benefits of higher resolution in terms of better topographic representation, and the issues of representation caused by the regional weather phenomenon and station density (Faiz et al., 2018; Isotta et al., 2015). Source-specific assessments showed that reanalysis datasets failed to represent quite a lot of weather patterns (Bosilovich et al., 2008), and satellite observation-based datasets were found restricted under cloud-covered conditions (Kidd et al., 2012).

Hence, the inherent issues in GPDs (Sun et al., 2018; Yao et al., 2020) make the selection of an appropriate input dataset very hard for specific hydrological applications, and thus, priori-evaluation of GPDs is necessary for hydrological simulation. Evaluation of the GPDs is performed generally with regard to the following aspects, intra- and inter-annual variation patterns (Yao et al., 2020) and water balance at multiple temporal and spatial scales. For a glacierized watershed, mass gaining from the precipitation or melt release as additional water input to streamflow (Shafeeque et al., 2019) makes the water balance

check more complicated.

The temporal distribution patterns of the GPDs are usually evaluated against the rain gauge data series (Ali et al., 2017; Anjum et al., 2018; Blacutt et al., 2015; Henn et al., 2018; Hu and Chen, 2018; Hu et al., 2016; Hussain et al., 2017; Romilly and Gebremichael, 2011). This is the most popular and widely accepted evaluation approach reported in open publications. Arguments also arise from this evaluation approach. The observation data might have already been used in the rain gauge





based GPDs. In this case, the data from corresponding grid cells may match the rain gauge data very well. On the other hand, this approach cannot be performed in the high elevation zone, where the rain gauges are usually unavailable. In fact, the station observations cannot capture the strong gradients in multiple local factors in mountainous regions, and that makes it hard to analyze and understand the spatiotemporal variations in hydroclimatic regimes (Lutz et al., 2014). The mountain areas are facing challenges in observed precipitation, primarily due to a small number and unevenly distributed

meteorological stations (Salio et al., 2015). Moreover, low altitude meteorological stations cannot represent a significant amount of precipitation at higher elevations (Dahri et al., 2018), which is expected to have a strong effect on the complex glacio-hydrometeorological systems. Besides, the limited gauge numbers usually are not enough for evaluating the GPDs with regard to the watershed water balance.

Using the available GPDs to drive the hydrological simulation is another way of evaluating the GPDs in both temporal-

spatial distributions and water balance (Tong et al., 2014). The detailed hydrological simulation outputs provide rich information for evaluating the GPDs and diagnosing potential problems. This is the most powerful way, yet the most complicated task to fulfill. It requires the specific knowledge of hydrological model and application, simulation setup on the basis of a huge amount of watershed simulation data, model parameterization (Ragettli and Pellicciotti, 2012), and capacity for output analysis. When this approach is applied to a glacierized watershed, knowledge of glacier changes, and glacio-

hydrological processes are required. Data scarcity in the glacierized regions, changes in glacier storage, and glacial melt as an additional input to streamflow (Wang et al., 2018) make the GPD evaluation complicated.

Although the discharge measurements are more authentic than precipitation in snow-fed or glacierized river basins (Henn et al., 2015; Kretzschmar et al., 2015) and used to evaluate the watershed balance in a direct way, studies indicated that observed precipitation is insufficient to sustain the hydrological and glacier systems in mountain regions (Immerzeel et al.,

2015), and thus, cannot represent the real amount of precipitation in a glacierized river basin as well.

Glacier mass change is a key issue in checking the performance of GPDs via the watershed water balance. On the one hand, part of precipitation may fall as snow and, part of the snow may turn into glacier ice eventually. The stored precipitation in solid-state moves with glacier and may melt at its lower end with time delay over the years (Cuffey and Paterson, 2010). The meltwater goes into streamflow as an additional water source (Shafeeque et al., 2019). The state change of precipitation ice -

meltwater can only be evaluated through the glacio-hydrological simulations, which is usually a complicated task to do.

Truc-Budyko theory (Budyko, 1974; Truc, 1954) deals with water and energy at the watershed and annual scale (Coron et al., 2015; Valéry et al., 2010). It connects the annual streamflow to the precipitation and evapotranspiration in a formula, and thus, provides a simple method to evaluate the annual precipitation by the observed streamflow. It is usually applied with an assumption that the annual change in watershed storage is neglected.

Provided that glacial storage change has involved in the water balance, it can be detected by the Truc-Budyko theory. When watershed water storage change is neglected, (a) $Q$ must not be negative (i.e., $Q \geq 0$), (b) $Q$ must not be less than the difference between precipitation and potential evapotranspiration (i.e., $Q \geq P - ET_p$), and (c) $Q$ must be less than the available precipitation (i.e., $Q \leq P$) (Andréassian and Perrin, 2012). When glacier storage changes and meltwater releases to





streamflow, the relationship among precipitation, evapotranspiration, and streamflow will go beyond the principles
mentioned above. The potential problems of GPDs and the involvement of glaciers can thus be detected.

 A Tri-approach is proposed in the current study, which is a combination of (1) statistical validation—comparisons of GPDs
against the observed precipitation based on climatology (2) physical diagnosis—assessing physical realism of GPDs to
represent a plausible water-energy balance, and (3) practical simulation—based on the simulation of hydrology and glacier
changes using glacio-hydrological models. The purpose of this framework is to provide a way of evaluating a GPD from
multiple perspectives, diagnose the potential problems in it, and suggest the directions for its local correction.

The Tri-approach is applied as a case study in the glacierized Upper Indus Basin (UIB), which is located in a high elevation
zone and covered by extensive glaciers (Bajracharya and Shrestha, 2011).

Some popular GPDs, such as APHRODITE, CFSR, HAR, PGMFD, and TRMM, are evaluated in UIB using the proposed
Tri-approach as an example application. Performance of these datasets is evaluated by answering: (a) How do the GPDs
perform against the observed precipitation? (b) Can the GPDs represent the real water-energy balance (physical realism)?
and (c) Can GPDs simulate the rational outputs of hydrology and glacier changes in glacierized catchments simultaneously?
Based on the evaluation, suggestions to further corrections to these datasets are made.

## 2 Materials and Methods

### 2.1 Study area

Upper Indus Basin (UIB) covers an area of more than 173,000 km$^2$ shared among China, India, and Pakistan (31−37°N and
72−82°E). About 50% area of UIB lies within Pakistan. UIB hosts the eastern Hindukush, western Himalaya, and Karakoram
mountain ranges (Inman, 2010; Khan et al., 2015; Mukhopadhyay and Khan, 2014). The hydroclimatic characteristics of
different spatial sub-regions of UIB are different from each other. Westerlies and summer monsoon precipitation systems
(Figure 1a) are responsible for the annual precipitation in UIB; however, the effect and contributions of both sources differ
temporally and spatially. The total number of glaciers is about 12,000, having more than 15,000 km$^2$ glacier area with glacier
area ratio (GAR) of about 9% (Bajracharya and Shrestha, 2011) (Figure 1c). The percent snow-covered area of UIB varies
from more than 10 to 70% (Hasson et al., 2014), In UIB, the snow cover has distinct seasonal patterns with maximum snow
in spring and a minimum in summer (Gurung et al., 2017). It is difficult to treat the UIB as a single unit because of the
influence of multiple climatic systems as well as unique interactions among the cryosphere, atmosphere, and hydrosphere
(Palazzi et al., 2013). Therefore, in this study, UIB was divided into three sub-regions based on their spatial location in the
three major mountain ranges: Himalaya, Hindukush, and Karakorum ([ and Figure 1b).

[Table 1]

[Figure 1]



## 2.2 Data collection and preparation

The details of hydrology, climate, soil, land use, DEM, glacier mass balance, and reanalysis datasets are provided in [. Annual average observed discharge varied between 140–2431 m³/sec at the given hydrological stations (Figure 1e). The average annual temperature was 5.3 ºC in the entire domain of UIB for 1985−2014. The average annual minimum and maximum temperature was -0.98 ºC and 11.58 ºC, respectively (Supplementary Fig. S1).

**[Table 2]**

The basic wrangling, analysis, and extraction of GPDs were done using Climate Data Operators (CDO v1.9.7) package (https://code.mpimet.mpg.de/projects/cdo), GIS 10.2, and R (https://www.r-project.org). The spatial fields for adjusted observed precipitation (OBS) were generated by interpolating and resampling the point observations. The OBS and GPDs were resampled at a common resolution of 0.25º×0.25º. A simple resampling technique 'nearest neighbor' was used to resample the datasets at 0.25º×0.25º resolution.

### 2.2.1 Evapotranspiration calculation

The $ET_p$ was calculated based on the Hargreaves method (Hargreaves and Samani, 1985) using Drought Indices Calculator (DrinC V1.7) software (Tigkas et al., 2014). The calculated $ET_p$ was between 795 mm yr⁻¹ to 1015 mm yr⁻¹, with an average of 907 mm yr⁻¹ ([). The highest values were in the Hindukush, while the lowest in the Karakorum sub-region.

The choice of the formula is very critical in calculating $ET_p$ (Zhou et al., 2020) because it can affect the rest of the analysis; therefore, it is essential to validate calculated $ET_p$. For validating the calculated $ET_p$ and calculating the actual evapotranspiration ($ET_a$), reference $ET_p$ data (Supplementary Fig. S2a) was also extracted from Global Reference Evapo-Transpiration (Global-ET0) at 1km resolution over the period 1970−2000 (Antonio and Zomer, 2018), available at the CGIAR-CSI GeoPortal. In this product, the potential evapotranspiration data was estimated based on the FAO Penman-Monteith method. The actual evapotranspiration data (Supplementary Fig. S2b) was extracted from Esri_hydro "average annual actual evapotranspiration" derived by the researchers at the University of Montana based on the data from MOD16 Global Evapotranspiration Product (ESRI, 2019). The $ET_a$ was estimated using the following relationship:

$$ET_a = \left( {ET_{a_M}}\Big/{ET_{p_M}} \right) \times ET_p \tag{1}$$

Here, $ET_a$ represents the estimated actual evapotranspiration, $ET_{a_M}$ is the actual evapotranspiration based on Esri_hydro "average annual actual evapotranspiration", $ET_{p_M}$ is the potential evapotranspiration based on Global Reference Evapo-Transpiration, and $ET_p$ is calculated evapotranspiration. The $ET_a$ was less than $ET_{a_M}$ by -17% to -4% in sub-regions and -averagely -11% in entire UIB ([).

**[Figure 2]**





### 2.2.2 Precipitation adjustment

Observed precipitation encounters with several uncertainties including undercatch snow, wind effect on snow redistribution,
catchment located on the leeward side (e.g., Himalaya sub-region), low station density, uneven distribution of observation
network, and variations in overlapping periods for all the meteorological stations. The problem of low density and uneven
distribution of meteorological stations is huge, common, and unavoidable in most of the alpine regions (Isotta et al., 2015;
Liu et al., 2019), while the undercatch and measurement errors may also be amplified during different seasons (Rasmussen et
al., 2012). Furthermore, the elevation of UIB ranges between ~300−8569 masl; however, the meteorological stations are
located below ~5000 masl with an average elevation of 3100 m (Figure 1b). Hence, there are no observed data at above 5000
masl, which is an unavoidable limitation in this region (Winiger et al., 2005). Previous studies (Basist et al., 1994;
Bookhagen and Burbank, 2006; Hu et al., 2015; Immerzeel et al., 2015; Johansson and Chen, 2003; Yoon et al., 2019) have
proven that precipitation is largely affected by topography, and this correlation is due to vertical deflection of moist winds
aloft, the hindrance or modification of low pressure and frontal systems, and the promotion of local convection currents (Roe,
2005). So, the low elevation meteorological stations might not represent the higher elevation precipitation; therefore, using
such station data for performance evaluation may induce uncertainty in the analyzed results. A number of previous studies in
the region have corrected the precipitation using different reverse modeling approaches (Dahri et al., 2018; Immerzeel et al.,
2015). In the current study, observed precipitation is adjusted for the selected station based on the corrected precipitation in
Dahri et al. (2018). The interpolated observed precipitation was resampled at 0.25º×0.25º resolution based on the 'nearest
neighbor resampling technique' for comparison purposes. The adjusted precipitation was higher by 73% than the uncorrected
observed precipitation in UIB ([a). The average annual adjusted precipitation was 540±180 mm yr$^{-1}$ from 1951−2017 in UIB.
The spatial distribution of adjusted precipitation in UIB is presented in [b.

**[Figure 3]**

### 2.3 The Tri-Approach framework

The Tri-approach framework includes three approaches: (a) statistical performance evaluation of GPDs against OBS for
investigating the ability of GPDs to represent the climatology, (b) testing the physical realism of GPDs to represent the
plausible water-energy balance based on a hydrological alternative of Truc-Budyko theory, and (c) practical simulation using
hydrological models to investigate the rationality of simulated hydrology and glacier changes. A schematic diagram of Tri-
approach is provided in [a.

185                                                    **[Figure 4]**

### 2.3.1 Statistical analysis

The statistical analysis is based on the comparisons between GPDs and OBS precipitation using different statistical indices.
The analysis was performed to identify the differences among the GPDs in representing the patterns of monthly and seasonal
distribution and inter-annual variations of OBS precipitation. The statistical indices include correlation coefficient (CC) ((2),





percent bias (PBIAS) ((3), root means square error (RMSE) ( (4), and standard deviation (SD) ((5). The similarity in spatial

or temporal patterns between two datasets can be indicated by CC, the absolute mean difference between two datasets can be

measured by RMSE, the systematic under- or over-estimation of a dataset can be shown by PBIAS. SD can measure the

spread of data about the mean value. All of these statistical parameters were calculated as follows:

$$CC = \frac{\Sigma_1^n\left(P_i^{obs} - P_{mean}^{obs}\right)\left(P_i^{gd} - P_{mean}^{gd}\right)}{\sqrt{\Sigma_1^n\left(P_i^{obs} - P_{mean}^{obs}\right)^2}\sqrt{\Sigma_1^n\left(P_i^{gd} - P_{mean}^{gd}\right)^2}} \tag{2}$$

$$PBIAS = \frac{\Sigma_1^n\left(P_i^{gd} - P_i^{obs}\right)}{\Sigma_1^n\left(P_i^{obs}\right)} \times 100 \tag{3}$$

$$RMSE = \sqrt{\frac{\Sigma_1^n\left(P_i^{obs} - P_i^{gd}\right)^2}{n}} \tag{4}$$

$$SD = \sqrt{\frac{1}{n-1}\Sigma_{i=1}^n(P_i - \bar{P})^2} \tag{5}$$

Here, $P_i$ is the $i^{th}$ observation for the precipitation (superscript '*obs*' and '*gd*' represents OBS and GPD, respectively), $P_{mean}^{obs}$

is the mean of observed precipitation, $P_{mean}^{gd}$ is the mean value for the precipitation in GPDs, and *n* is the total number of

values in the corresponding dataset.

The precipitation data was available varying from 1951 to 2017 for different datasets ([) and OBS precipitation

(Supplementary Table S1). The climatology was derived for Himalaya, Hindukush, and Karakorum sub-regions using all the

available data for each rescaled dataset. The spatial distribution of GPD and OBS precipitation was explored based on annual

average precipitation in sub-regions of UIB. Temporal distribution of OBS and GPDs precipitation was explored at monthly,

seasonal, and annual time scales in sub-regions of UIB. The trend analysis of the OBS and GPDs was performed based on

the Mann-Kendall test, which has been widely used for non-parametric analysis in hydrometeorological studies (Hirsch et al.,

1991). Sen's non-parametric method (Sen, 1968) was used to estimate the slope value (Supplementary Information). The

annual cycle was divided into four seasons: (a) March, April, and May (MAM)—Spring, (b) June, July, and August (JJA)—

Summer, (c) September, October, and November (SON)—Autumn, and (d) December, January, and February (DJF)—

Winter. It is important to mention that in some of the previous studies, only two seasons were considered to represent the

seasonal precipitation, i.e., winter (usually Oct−Mar) and summer (Jul–Sep) season, e.g., (Dahri et al., 2016; Hewitt, 2007).

The annual average precipitation for each dataset was compared with OBS precipitation for the corresponding overlapped

period in sub-regions of UIB. Taylor's diagram (Taylor, 2001) was used to express the comparison results graphically.

ETCCDI indices (Peterson, 2005) are generally used to analyze the extreme precipitation characteristics in GPDs (Nastos et

al., 2013). In this study, four ETCCDI, including consecutive dry days (CDD), consecutive wet days (CWD), precipitation



due to extremely wet days (R99pTOT), and simple precipitation intensity index (SDII), were selected to compare the performance of GPDs in representing the precipitation extremes. CDD is the maximum length of dry spells with precipitation <1mm, CWD is the maximum length of wet spells with precipitation >1mm, R99pTOT is the annual total

precipitation when daily wet day amount >99[th] percentile, and SDII is the mean precipitation amount on wet days. RClimDex software package (https://github.com/ECCC-CDAS/RClimDex) was used to calculate ETCCDI. The average values of ETCCDI for GPDs were compared with those for OBS in sub-regions of UIB.

### 2.3.2 Physical diagnosis

The physical diagnosis was performed to identify the actual over- and under-estimation of the GPDs at watershed and annual

scales based on the water-energy balance of the watershed. The hydrological alternative of the Truc-Budyko plot was used to diagnose the GPD for reproducing a plausible water-energy balance. First, the water-input was compared with the water-output in the sub-regions of UIB. To do so, the precipitation from selected datasets was compared with the specific runoff to assess the quantitative relationship between the specific runoff and precipitation in different datasets, including OBS at monthly and annual scales. Then, a non-dimensional representation of physical water-energy balance was applied to estimate

the actual under- or over-estimation in the glacierized catchment. The most widely used type of representation is proposed by Truc (1954) and Budyko (1974). Finally, the water balance equation of a glacierized catchment was used to estimate the change in glacier storage for each dataset.

In this study, the physical realism of each precipitation dataset was verified using a hydrological alternative of a non-dimensional Truc-Budyko plot (Andréassian and Perrin, 2012). In this approach, the realistic closure of water-energy

balance was tested using precipitation from each dataset. Long-term water yield or runoff coefficient ($Q/P$) was plotted as a function of long-term aridity-index ($P/ET_p$) (Coron et al., 2015; Valéry et al., 2010), i.e.,

$$Q/P = f\left(P/ET_p\right) \tag{6}$$

Here, $Q$, $P$, and $ET_p$ represent specific runoff, precipitation, and evapotranspiration in a catchment, respectively. Plotting aridity-index on the x-axis also allows focusing on the wettest and driest catchments based on input precipitation (wetter catchment corresponds to higher $P/ET_p$ value). The physical interpretation of this hydrological representation is based on

three assumptions: (1) $Q \geq 0$, (2) $Q \geq P - ET_p$, and (3) $Q \leq P$ ([b). All three limits are based on the water balance equation of a water-tight (conservative) catchment. The water balance equation for a conservative catchment can be written as follows:

$$P = Q + \alpha ET_p \qquad \text{(where } \alpha \leq 1) \tag{7}$$

The point (representing a catchment—different positions in the plot for different precipitation datasets), which falls within the feasible domain is considered as realistic or "True" catchment ([b). The feasible domain is an area below or equal to the





water limit and above or equal to the energy limit. If a data point falls above the water limit (i.e., $Q > P$) or below the energy
limit, then it is called "Gaining" or "Leaky" catchment, respectively (Andréassian and Perrin, 2012).

When a glacierized catchment falls in the "Gaining" zone ($Q > P$) ([b) based on the water-energy balance, it implies that
there must be an additional water term that contributes to total runoff. The meltwater contributions to total runoff highlight
that higher precipitation is required to sustain such glacier systems in glacierized catchments. Hence, the precipitation in that
GPD is underestimated as compared to the actual water-input in a glacierized catchment. In the case of "Leaky" catchment
(Fig. 2b), when the runoff is less than the available energy, it implies that a part of total runoff is missing from the water
balance or the precipitation is overestimated. In glacierized catchments, the missing water can be stored in the form of a
positive glacier mass balance. Therefore, playing with the rational output of mass balance and streamflow can lead to the
corrected precipitation, which would be sufficient for sustaining both water and mass balance.

In glacierized catchments, the simplest water balance equation can be written as:

$$\Delta S/\Delta t = P - (Q + ET_a) + MB \tag{8}$$

Here, $ET_a$ and $MB$ represent actual evapotranspiration and mass balance in the watershed, respectively. The imbalance in (8
is the change in storage ($\Delta S$). When $\Delta S = 0$, the catchment is the perfect "True" catchment. The "True" catchment can have a
slight positive or negative change in storage depending on the quality of observed mass balance and $ET_a$. However, the
"Gaining" catchments always have a negative change in storage ($\Delta S < 0$), and the "Leaky" catchments always have positive
changes in storage ($\Delta S > 0$). The negative change in storage results in the melting of glaciers and contributing additional
water to total runoff. Whereas, positive change in storage represents advancing glaciers in a catchment where heavy
precipitation falls in solid form and stored in the form of glaciers.

### 2.3.3 Practical simulation

The practical simulation is used to ensure that GPDs are capable of producing a balanced output of streamflow and glacier
changes at the same time. The observed glacier, snow cover, and hydrology data are used to calibrate the glacio-hydrological
model, and then the simulated results of runoff and mass balance are analyzed for rationality in a glacierized catchment. The
Spatial Processes in HYdrology (SPHY) model (Terink et al., 2015) was used for the practical validation of the ability of all
the precipitation datasets to simulate hydrology and glacier changes. The SPHY model is a fully distributed, leaky bucket
type hydrologic model. It has been developed using key components of HydroS (Droogers and Immerzeel, 2010), PCR-
GLOPWB (Bierkens and van Beek, 2009), SWAT (Arnold et al., 1998), HimSim (Immerzeel et al., 2012), and SWAP (Van
Dam et al., 1997). The SPHY model is a raster-based glacio-hydrological model, and it has been used in different glacierized
regions. The primary advantage of the SPHY model is its glacier module, which can distinguish between the clean ice and
debris-covered glaciers. The debris-covered glacier area is ~18% (Khan et al., 2015), and debris-covered glaciers can affect





the overall meltwater contributions (Gardelle et al., 2012; Kaab et al., 2012; Khan et al., 2015). The SPHY model allows assigning different degree-day factors for the two to differentiate between their melt rates.

In the SPHY model, total runoff is a sum of four possible components: glacier runoff, snow runoff, baseflow runoff, and rain runoff.

$$R_{total} = R_{snow} + R_{glacier} + R_{rain} + R_{baseflow} \tag{9}$$

Here, $R$ represents the runoff (mm) for a unit time step. The glacier runoff is composed of supraglacial snowmelt, ice melt, and direct rain on ice runoff. As the SPHY model is based on grid cell spatial discretization, sub-grid parameterization is used to differentiate glacier cover, i.e., clean and debris-covered glacier fraction within a grid cell using a debris cover mask

starting from lower elevations. The dynamic snow and soil water storage are solved within the remaining fraction of the grid cell. The detailed description of the SPHY model is referred to Terink et al. (2015). Here, a brief description of snow and glacier runoff generation is provided.

Based on the threshold temperature, precipitation is differentiated into rain or snowfall ($P = Snow$, when $T_{avg} \leq T_{threshold}$). Snowmelt is calculated using degree-day model (Hock, 2003) as follows:

$$M_{snowPot} = \begin{cases} T_{avg} \times DDF_{snow} & ; when\ T_{avg} > 0 \\ 0 & ; when\ T_{avg} \leq 0 \end{cases} \tag{10}$$

$$M_{snowAct} = \min\left(M_{snowPot_t}, \Delta Snow_{t-1}\right) \tag{11}$$

Here, $M_{snowPot}$ is the potential snowmelt (mm), and the actual snowmelt ($M_{snowAct}$) (mm) is calculated using the snow storage of the previous day ($\Delta Snow_{t-1}$). $DDF_{snow}$ (mm °C$^{-1}$ day$^{-1}$) is the degree-day factor for snow, and it is a calibration parameter. The snow runoff is generated when the melting point is below the air temperature, and melted snow cannot be refrozen within the snowpack. The snow runoff is the balance of actual snowmelt, liquid precipitation, and the refrozen meltwater.

$$R_{snow} = M_{snowAct} + P_{liquid} - M_{refrozen} \tag{12}$$

The glacier melt is calculated by differentiating clean and debris-covered glaciers because both categories have different

melt rates (Reid et al., 2012). Degree-day model (Hock, 2003) was adopted to calculate the melt as follows:

$$M_{glacierCI} = \begin{cases} T_{avg} \times DDF_{glacierCI} \times F_{glacierCI} & ; when\ T_{avg} > 0 \\ 0 & ; when\ T_{avg} \leq 0 \end{cases} \tag{13}$$

$$M_{glacierDC} = \begin{cases} T_{avg} \times DDF_{glacierDC} \times F_{glacierDC} & ; when\ T_{avg} > 0 \\ 0 & ; when\ T_{avg} \leq 0 \end{cases} \tag{14}$$


Here, $M_{glacierCI}$ and $M_{glacierDC}$ is the daily glacier melt from clean ice and debris-covered glaciers, respectively; $F_{glacierCI}$ and $F_{glacierDC}$ is the fraction of debris-free and debris-covered glaciers, respectively; $DDF_{glacierCI}$ (mm °C$^{-1}$ day$^{-1}$) and $DDF_{glacierDC}$ (mm °C$^{-1}$ day$^{-1}$) is the degree-day factor for debris-free and debris-covered glaciers, respectively. The total glacier melt within a grid cell ($M_{glacierT}$) is calculated by multiplying the total glacier fraction ($F_{glacier}$) with the sum of daily glacier melt

from debris-free and debris-covered glaciers as follows:

$$M_{glacierT} = \left( M_{glacierCI} + M_{glacierDC} \right) \times F_{glacier} \tag{15}$$

The glacier runoff ($R_{glacier}$) is calculated as a product of glacier runoff factor ($G_{RF}$—a calibration parameter used to allow the percolation) multiplied by total glacier melt within a grid cell as follows:

$$R_{glacier} = M_{glacierT} \times G_{RF} \tag{16}$$

The remaining meltwater percolates into soil layers and recharges groundwater, which after baseflow recession days ($BF_{days}$—a calibration parameter), is added up in total runoff as baseflow (Terink et al., 2015).

The model was forced for 2001−2007 with a one-year warm-up period at a daily time step and spatial resolution of 1-km. Initially, the model was calibrated and validated for OBS precipitation over 2002−2004 and 2005−2007 in UIB, respectively. A three-fold multi-objective calibration is adopted to avoid the issues of equifinality caused by the glacier compensation effect.

In the first step, the degree-day factors for clean ice and debris-covered glaciers were optimized ([) based on the area-

weighted mean glacier mass balance. The observed mass balance data were extracted from the literature. The mass balance in the SPHY model was taken as the accumulation in the form of solid precipitation on the grid cell with glacier fraction and adjacent grid cells with a slope steeper than 0.2 (Immerzeel et al., 2015). Then, the parameters related to snow were calibrated based on the snow extent in the basin. The average monthly snow cover was compared with MODIS snow cover, which was averaged over for every month from the MODIS 8-day product. In the third step, the parameters related to

baseflow and routing were calibrated based on observed daily runoff at Besham Qila gauge station.

After parameterizing the base SPHY project, six SPHY projects were developed using the calibrated set of sensitive parameters for each precipitation dataset in the entire domain of UIB. All SPHY projects had the same datasets and all other specifications, except precipitation data. To assess the rationality of simulated glacio-hydrological results, either runoff or the mass balance should be identical among all the SPHY projects. Therefore, the SPHY projects based on GPDs were re-

tuned for glacier and snow parameters to achieve a similar average mass balance for all the SPHY projects. The rationality between the glacio-hydrological outputs was investigated. Comparisons among the simulated glacio-hydrology for each SPHY project were made for hydrological performance at daily scale. The inter-annual variations in total runoff and mass balance and PBAIS with observed runoff were investigated. The contributions of total runoff components were also compared among the outputs of six SPHY projects.



315                                                        **[Table 3]**

## 3 Results

### 3.1 Statistical validation based on climatology

The first component in the Tri-approach framework is the statistical comparisons among the abilities of GPDs in representing the climatology of OBS precipitation. The GPDs were compared against OBS over the varying periods from
1951 to 2017 to evaluate their spatiotemporal performance. In all the GPDs, the northeast part of UIB had low precipitation compared to the other parts, whereas the southwest part with minimum average elevation had the maximum amount of precipitation ([a). Spatial distribution patterns in CFSR and HAR were in resemblance with that of OBS; however, the amount of precipitation was overestimated. In UIB, average annual precipitation was found 323±99 mm yr$^{-1}$, 1115±419 mm yr$^{-1}$, 955±218 mm yr$^{-1}$, 410±84 mm yr$^{-1}$, and 342±86 mm yr$^{-1}$ for APHRO, CFSR, HAR, PGMFD, and TRMM over varying
periods from 1951 to 2017, respectively. The trend analysis of OBS precipitation data showed a significant positive trend in all sub-regions, except the Hindukush (Supplementary Table S3). Mann-Kendall test statistics revealed that all the GPDs showed random trends in UIB.

**[Figure 5]**

The evaluation results of the GPDs are graphically presented using Taylor's diagrams in sub-regions and UIB ([. The
proximal distance and position on correlation bars represent the performance. The CC for CFSR was the lowest (<0.2), whereas the highest values of bias (106%) in UIB ([. The performance of HAR to represent the inter-annual variations of OBS was also unsatisfactory due to higher bias (77%) and lower correlation. The performance of APHRO to represent the pattern of annual variations was identified as the better in UIB with a higher correlation (CC > 0.6) ([; however, it was underestimated by -50% in UIB ([.

335                                                      **[Figure 6]**

**[Table 4]**

The annual precipitation cycle is represented using the hyetographs for each dataset in three sub-regions at the monthly time scale ([a). The annual cycle of OBS precipitation had a bi-modal hyetograph, where the first peak occurred in April and second in August in all sub-regions. The monthly distribution of area-weighted precipitation indicated a bi-modal weather
system in UIB. The annual precipitation distribution pattern was associated with the westerlies and Indian monsoon in winter and summer, respectively. The first peak of the OBS precipitation is due to the westerlies as most of the precipitation occurs in the winter and spring seasons in solid form. On the other hand, the second peak is due to the summer monsoons in the region. Most datasets capture the second peak of OBS precipitation in the Himalaya sub-region, while the first peak was mimicked by most of the datasets in Hindukush and Karakorum sub-regions. This highlights that the GPDs can represent the
influence of westerlies up to some extent in Karakorum and Hindukush sub-regions, and monsoon in Himalaya. However, the precipitation amount is underestimated in APHRO, TRMM, and PGMFD; whereas, it is overestimated in CFSR and





HAR as compared to the OBS. APHRO performed better (CC > 0.8) in representing the patterns of monthly distribution of OBS (Supplementary Fig. S5).

The seasonal distribution of precipitation was explored and compared for the winter, spring, summer, and autumn seasons in
sub-regions of UIB ([b). The most part (61%) of annual OBS occurred in the winter and spring season in UIB. In sub-regions, Himalaya, Hindukush, and Karakorum, 63%, 62%, and 56% of annual precipitation occurred in winter and spring ([b). Some of the previous studies combined the two seasons and labeled it as winter precipitation. Averagely, the winter and spring season precipitation was overestimated by CFSR and HAR by 13% and 22%, respectively, whereas, it was underestimated by APHRO (-23%), PGMFD (-7%), and TRMM (-17%) as compared to the OBS in UIB. This highlights that mostly
westerlies influenced the distribution of annual precipitation in UIB.

**[Figure 7]**

The average values of precipitation extremes in OBS and GPDs based on selected ETCCDI are presented in [. CDD based on APHRO was the highest in Himalaya and Karakorum sub-region, while CDD based on TRMM was the highest in Hindukush. Among the GPDs, the lowest CDD values were for CFSR in all the sub-regions. Average values of CDD for
OBS, APHRO, CFSR, HAR, PGMFD, and TRMM were 30±20 days, 52±27 days, 22±6 days, 26±7 days, 46±25 days, and 45±18 days in all sub-regions, respectively. It showed that the average duration of dry spells CDDs in APHRO, PGMFD, and TRMM was overestimated, and it was underestimated in CFSR and HAR as compared to that in OBS. On the contrary, the maximum length of wet spells was found the longest for CFSR, while the lowest for APHRO. The average values for CWD for OBS, APHRO, CFSR, HAR, PGMFD, and TRMM were 10±4 days, 6±2 days, 22±13 days, 13±5 days, 8±3 days,
and 8±4 days in UIB, respectively. The highest value for R99pTOT was for OBS averagely, and all GPDs were underestimated for this ETCCDI compared to OBS. Among the GPDs, CFSR had the highest value for R99pTOT, whereas APHRO had the lowest. The SDII values for HAR, CFSR, and PGMFD were greater than the OBS, whereas APHRO and TRMM showed smaller values. The greatest SDII was for HAR dataset.

**[Figure 8]**

**3.2 Physical validation based on the water-energy balance**

**3.2.1 Precipitation versus specific runoff**

First, the monthly specific runoff was compared to the monthly area-weighted region-wise precipitation. It was found that the intra-annual distribution of specific runoff was quite different from that of precipitation ([). The comparisons among the datasets were made using the runoff coefficients (*Q/P*) for each month. The values of runoff coefficients greater than one
(*Q/P > 1*) means that runoff is higher than the precipitation. Based on the values of runoff coefficients, it was noted that the runoff peaks occurred during Dec-Apr in all sub-regions ([) because most of the precipitation fall in winter and spring seasons ([). The runoff coefficients in the Hindukush were lower than those in Himalaya and Karakorum sub-regions for all datasets. Such a relationship between the runoff and precipitation is because the winter and spring precipitation occurs mostly in solid form as snow and remains there until it starts melting and contributes to late-spring and early-summer flows.





The accumulation of snow during winter and melting of snow and glaciers during summer creates the difference between the distribution of precipitation and specific runoff.

**[Figure 9]**

Water-year precipitation totals based on all the selected datasets were compared with annual runoff in the Himalaya, Hindukush, and Karakorum over varying periods from 1983 to 2010, depending on the data availability and the overlapped

period of the respective dataset and sub-region ([]). Considerable differences were spotted among the datasets when compared with the runoff in sub-regions. The water-year total precipitation was identified lesser than the runoff in the Hindukush and Karakorum sub-region for APHRO, PGMFD, and TRMM. The correlation between annual runoff and precipitation was significant and satisfactory for APHRO, PGMFD, and TRMM in the Himalaya sub-region. Overall, TRMM and PGMFD showed good and significant correlation with annual runoff in UIB (0.68 and 0.54, respectively);

however, they were lesser as compared to the annual runoff. The impact of the Indian summer monsoon probably played a significant role in greater precipitation totals in the Himalaya sub-region.

**[Figure 10]**

### 3.2.2 Physical realism to reproduce plausible water balance

The physical realism of each dataset to represent the water balance in each region was tested and plotted based on a

hydrological alternative of the Truc-Budyko plot ([]). The aridity index ($P/ET_p$) and runoff coefficient ($Q/P$) were plotted on the x-axis and y-axis, respectively. Each point represents a catchment, and the colors differentiate among different datasets. The catchments fallen within the feasible domain were considered as physically realistic or "True" catchments. In "True" catchments, precipitation was enough to reproduce the water balance; however, this amount of precipitation may or may not be sufficient to represent the mass balance in glacierized catchments.

The points above $P/Q = 1$ line or under the energy limit (right side of theoretical Budyko line) were considered physically unrealistic. Most of the points were above the $Q/P = 1$ line for OBS and other gridded datasets except CFSR. These points represent the "Gaining" catchments. In gaining catchments, precipitation was not sufficient to close the water balance. The points out of the energy limit (i.e., $Q < P-ET_p$) were characterized as "Leaky" catchments, i.e., runoff deficit was greater than the potential evapotranspiration, in our case, HAR and CFSR represent the Hindukush, Karakorum, and average of entire

UIB as "Leaky" ([]). Such behavior and possible deviations can be explained by potential errors and uncertainties in observed runoff and calculated $ET_p$ in the study area. Moreover, the theoretical Budyko curve (energy limit) is usually different for glacierized basins because of an additional water term in water balance from glacier melting. Based on the aridity-index values, CFSR and HAR were identified to make the whole study area extremely wet.

In the "Gaining" catchments, which break the water limit ($Q > P$), additional water term is added to the water balance. This

additional water is contributed by glacier melt in the glacierized catchments, and such behavior of the catchment results in a negative change in glacier storage $(\Delta S < 0)$. For example, in the Himalaya sub-region, all the datasets except CFSR were "Gaining", and meltwater contributed to the total runoff. On the other hand, in the "Leaky" catchments, which break the





energy limit ($Q < P\text{-}ET_p$), some quantity of water is missing in the water balance. This missing water is stored in the form of positive glacier storage ($\Delta S < 0$). For example, CFSR and HAR made Hindukush and Karakorum sub-regions and entire UIB

domain as "Leaky", where missing water from the water balance may result in advancing glaciers.

[Figure 11]

The physical diagnosis modified from the Truc-Budyko theory with the addition of mass balance can provide quantitative information of change in glacier storage for different precipitation datasets. The change in glacier storage based on OBS, APHRO, CFSR, HAR, PGMFD, and TRMM was ranging between -0.20 m w.e. y$^{-1}$, -0.42 m w.e. y$^{-1}$, 0.37 m w.e. y$^{-1}$, 0.21 m

w.e. y$^{-1}$, -0.34 m w.e. y$^{-1}$, -0.40 m w.e. y$^{-1}$ in UIB, respectively ([). CFSR and HAR showed a positive change in storage for all the regions except Himalaya, where all datasets generated negative changes in storage. In Himalaya, CFSR resulted in a slightly negative change in storage, i.e., -0.06 m w.e. y$^{-1}$, which is because it was in the feasible domain. It is important to note that the datasets which represent "Gaining" catchments showed negative changes, i.e., OBS, APHRO, TRMM, PGMFD in storage because glacier melt contributed to total runoff in these catchments, resulting in the mass loss. CFSR and HAR

datasets represent "Leaky" catchment behavior for Hindukush and Karakorum sub-regions as well as entire UIB ([ and [), and thus, resulted in positive glacier storage change because of overestimated precipitation.

[Figure 12]

**3.3 Practical validation based on simulated hydrology and glacier changes**

This section provides the results of glacio-hydrological simulations based on the SPHY model to testify the ability of GPDs

to generate the rational output of streamflow and glacier changes. These results may find the problems of temporal distribution, water balance, and involvement of glaciers, as found in previous sections.

The degree-day factors for debris-covered and debris-free glaciers were calibrated based on the observed mass balance data. The degree-day factor for snow, water storage capacity, and threshold temperature were calibrated using MODIS snow cover data in UIB. The baseflow and routing related parameters were calibrated using observed runoff data at Besham Qila

hydrological gauge station. The calibrated degree-day factors ([) fall within the range of observed degree-day factors in the Karakorum mountains (Zhang et al., 2006).

The average simulated mass balance was -0.17 m w.e. yr$^{-1}$ ([), which was in a very good agreement with the mass balance derived in previous studies (Brun et al., 2017; Gardelle et al., 2013; Kääb et al., 2012; Kääb et al., 2015; Muhammad et al., 2019).


[Figure 13]

The calibration of the snow-related parameters was performed based on snow cover variations in UIB. An excellent performance ($R^2 = 0.93$) was achieved using the MODIS data as observed snow cover and compared with the simulated snow cover in UIB ([. The simulated snow cover based on the SPHY model varies between 23% and 74% of the total basin area over 2002−2007 in UIB. The simulated snow cover is in well match with the MODIS snow cover in the UIB (b).

Previous studies indicated the minimum snow cover as less than 10% using MODIS data (Hasson et al., 2014). The





difference in values with our study is due to the study area size and selected period for the evaluations. They included the Jhelum and Kabul basins in their evaluations, where minimum snow cover is used to reduce up to less than 5% of the total basin area. The simulated snow cover is in a good match with a previous study in the region (Lutz et al., 2016).

**[Figure 14]**

The SPHY model was calibrated and validated for the baseflow, routing, and surface runoff based on the observed streamflow data in the UIB. The calibration results (2002−2004) show an excellent performance ($R^2 = 0.89$) of the model at a daily time step ([a-b]). The validation results (2002−2004) highlight excellent performance as well ($R^2 = 0.91$) ([c-d] based on the criteria suggested by Moriasi et al. (2007). The simulated flows during calibration and validation were slightly overestimated by 3.9% and 3.8%, respectively.

**[Figure 15]**

Six SPHY projects were set up in UIB using similar datasets except for precipitation, while one precipitation dataset was used to force one project. The annual simulated total runoff was underestimated for all the datasets except CFSR and HAR in UIB from 2002−2007 ([a]). The values of the correlation coefficient for calibrated SPHY projects ranged from > 0.80 to 0.95 in UIB from 2002−2007 ([b]). The average annual total runoff was $467\pm42$ mm yr$^{-1}$, $263\pm23$ mm yr$^{-1}$, $743\pm71$ mm yr$^{-1}$,

$684\pm76$ mm yr$^{-1}$, $330\pm34$ mm yr$^{-1}$, $299\pm33$ mm yr$^{-1}$ for SPHY_OBS, SPHY_APHRO, SPHY_CFSR, SPHY_HAR, SPHY_PGMFD, and SPHY_TRMM from 2002−2007 in UIB, respectively ([c]). The average runoff in SPHY_CFSR and SPHY_HAR was greater than that of SPHY_OBS, whereas SPHY_OBS, and SPHY_PGMFD, and SPHY_TRMM had comparatively lower runoff. The PBIAS of annual runoff simulated by SPHY_OBS, SPHY_APHRO, SPHY_CFSR, SPHY_HAR, SPHY_PGMFD, and SPHY_TRMM with the observed annual specific runoff was 4%, -41%, 66%, 53%, -

26%, -33% from 2002−2007 in UIB, respectively. The highest positive PBIAS was noted for SPHY_CFSR followed by SPHY_HAR, whereas the maximum negative PBIAS was noticed for the SPHY_APHRO project ([d]).

**[Figure 16]**

To assess the rationality in the simulated glacio-hydrology in UIB for different precipitation datasets, the SPHY projects were calibrated to produce similar average mass balance as in the base calibrated model (i.e., SPHY_OBS). In UIB, the

simulated mass balance by SPHY_OBS, SPHY_APHRO, SPHY_CFSR, SPHY_HAR, SPHY_PGMFD, and SPHY_TRMM was -0.17±0.17 m w.e. y$^{-1}$, -0.17±0.21 m w.e. y$^{-1}$, -0.17±0.48 m w.e. y$^{-1}$, -0.17±0.56 m w.e. y$^{-1}$, -0.17±0.18 m w.e. y$^{-1}$, and -0.17±0.10 m w.e. y$^{-1}$ for 2002−2007, respectively ([e]). It highlights that when the simulated mass balance is calibrated for the observed mass balance in the basin, the simulated runoff breaks the rationality of glacio-hydrological outputs. In such cases, simulated runoff is either over- or under-estimated as compared to the observed runoff in the basin. In the current

study, SPHY_CFSR and SPHY_HAR simulated overestimated runoff, whereas SPHY_APHRO, SPHY_PGMFD, and SPHY_TRMM generated underestimated runoff as compared to the observed runoff in UIB ([d]).

In UIB, the total runoff was contributed first by snow runoff in the late spring to early summer, and then glacier runoff started contributing to generate maximum flows in summer. Meanwhile, summer monsoon also played a role in producing





peak values of total runoff during summer ([). Baseflow joined the total runoff having a recession of more than three and a

half months ([) after percolation during the melting season, in addition to the running baseflow runoff. At annual scale, the glacier runoff, snow runoff, baseflow runoff, and rainfall-runoff contributed to total runoff ranging between 44−49%, 30−35%, 14−20%, and 3−5%, respectively, simulated based on six SPHY projects, which were forced using OBS and GPDs ([a).

For all the SPHY projects, the snow runoff contributions were the highest in the spring season, while the glacier

contributions were highest in the summer. In the spring season, the snow runoff contributions were 48%, 64%, 56%, 59%, 65%, and 67% to the total runoff simulated under SPHY_OBS, SPHY_APHRO, SPHY_CFSR, SPHY_HAR, SPHY_PGMFD, and SPHY_TRMM during 2002-2007 over UIB, respectively ([b). It was noted that all the simulated snow runoff contributions based on GPDs were higher than those by SPHY_OBS. Similarly, in the summer season, for SPHY_OBS, SPHY_APHRO, SPHY_CFSR, SPHY_HAR, SPHY_PGMFD, and SPHY_TRMM, 51%, 62%, 59%, 55%,

61%, and 58% runoff was contributed by glacier runoff in UIB, respectively ([b). Again, the simulated glacier runoff contributions during the summer season were higher for the GPDs compared to the OBS. It is also important to mention that the combined amount of contributed water by glacier and snow runoff during summer and spring season were higher for CFSR (69%) and HAR (53%) as compared to OBS, whereas these were lower in the case of APHRO (-44%), PGMFD (-27%), and TRMM (-35%) ([b). It also highlights the irrational behavior of hydrological outputs as contributions from

meltwater were overestimated for overestimated GPDs and underestimated for underestimated GPDs while keeping the mass balance constant.

**[Figure 17]**

## 4 Discussions

In this study, a Tri-approach framework is proposed to diagnose the potential issues in GPDs from multiple perspectives.

This framework can identify the actual under- or over-estimation of GPDs on the basis of watershed water and energy balance, diagnose their possible causes, and provides directions for local correction. The approach was applied in UIB as a case study. It has the ability to investigate climatology, water-energy balance, and rationality of simulated hydrology and glacier changes in a mountain glacierized watershed.

### 4.1 Ability to diagnose the problems in representing climatology

The statistical analysis component in the Tri-approach framework basically helps to investigate the performance of GPDs in representing the observed climatology. This component focuses on the monthly and seasonal distribution and inter-annual variations and precipitation extremes in GPDs. The comprehensive diagnosis of GPDs in representing observed climatology would be very useful for temporal correction of GPDs and analyzing the expected uncertainties in simulated glacio-hydrologic outputs. This statistical approach is more common and has been applied in multiple previous studies to evaluate





the performance of GPDs. However, the authenticity of such statistical evaluation is questionable when the observed data is insufficient or of inferior quality due to the uneven distribution of meteorological stations, which is the case in high elevation glacierized river basin. In the current study, the observed precipitation data were adjusted using the corrected precipitation in UIB. The adjusted precipitation is 73% greater than the uncorrected precipitation.

The GPDs have differences in their spatiotemporal resolutions, covered time span, and underlying methodologies (Sun et al., 2018); therefore, in this study, the GPDs based on various sources and different methods were selected for analysis (i.e., reanalysis—CFSR, observed interpolation—APHRO, the combination of reanalysis and observed interpolations—PGMFD, satellite observations—TRMM, and downscaled model output—HAR).

The spatial and temporal distribution of mean annual GPDs' precipitation shows diverse differences in magnitudes and patterns when compared with OBS ([-8). APHRO performed better to represent the patterns in interannual variations (CC > 0.6) ([); however, it was highly underestimated (-40%) ([). The bimodal annual cycle of OBS precipitation ([) indicates a multi-sourced weather system in UIB, which is influenced by westerlies during the Winter and Spring seasons, whereas monsoon impacts the distribution during the summer season. In previous studies, researchers have explained the bimodal weather system in this region (Dahri et al., 2016; Hasson et al., 2017). UIB receives >60% of total annual precipitation during the winter and spring seasons ([b), which is in good agreement with the arguments of Hewitt (2007) that 67% of total annual precipitation occurs during the winter (Oct-Mar) in this region. The winter and spring precipitation were underestimated in APHRO (-23%), TRMM (-17%), PGMFD (-7%), whereas it was overestimated in HAR (22%) and CFSR (13%). The duration of dry spells (CDD) greater for APHRO, PGMFD, and TRMM and shorter for CFSR and HAR than the OBS, whereas the opposite was true for wet spells (CWD) in UIB. CFSR and HAR had the highest values for R99pTOT among the GPDs, while SDII was the heist for HAR. APHRO showed the lowest values for R99pTOT and SDII among the GPDs in UIB ([). Overall, the reanalysis GPDs had higher values for maxima of precipitation extremes, whereas the opposite was true for the interpolation-based and satellite GPDs in glacierized UIB. This is in good agreement with the conclusions of Nastos et al. (2013) that satellite dataset TRMM underestimates the R99pTOT at higher elevation regions.

Reanalysis datasets, i.e., CFSR and HAR, show overestimation in all the sub-regions, whereas observation-based APHRO and satellite-based TRMM show underestimations ([-7). Similar concluding remarks were made by different researchers for reanalysis, interpolated, and satellite-based datasets, for example, Liu et al. (2018), Liu et al. (2018), Yao et al. (2020), Ji et al. (2020), and Dahri et al. (2016). The reanalysis datasets account for both solid and liquid precipitation more consistently, which may explain their overestimation in high mountain glacierized regions (Blacutt et al., 2015), whereas observation interpolated and satellite estimations based datasets have difficulties in detecting the snowfall (Rasmussen et al., 2012; Wang et al., 2013). It is important to note that continuous biases and changes in both models and observing systems can introduce fake trends and variability into reanalysis outputs (Bengtsson, 2004); therefore, trends and variabilities from reanalysis datasets should be treated carefully for hydrological applications.

Although GPDs have captured the monthly distribution patterns of OBS precipitation ([), these datasets show significant differences in their monthly, seasonal, and annual magnitudes ([). Moreover, the large under- and over-estimations for the





gridded datasets over elevational profile may have been caused by the dynamic climatic system (Pang et al., 2014),
precipitation dependency on altitude (Immerzeel et al., 2015; Wortmann et al., 2018) and the approaches used to generate
these datasets (Harris et al., 2014; Huffman et al., 2010; Saha et al., 2010). The reason for the better representation of OBS
climatology by the APHRO dataset is the use of observed data in its generation; however, the precipitation at ungauged
elevation ranges is not extrapolated in APHRO dataset (Ji et al., 2020; Yatagai et al., 2012), which would affect its
application in mountain glacierized catchments.

It must be noted that these findings over the selected glacierized mountain sub-regions may allow for a performance
assessment of the presented datasets in general for glacierized alpine regions. It is essential to highlight that most datasets are
not independent of each other, as most of them include the same station observations directly or assimilate them in some way.
This is, however, a common problem in comparison studies, which cannot be avoided.

**4.2 Ability to diagnose the problems in representing the water-energy balance**

The introduction of the Truc-Budyko theory into the Tri-approach framework is useful to identify the actual under- or over-
estimation of GPDs on the basis of watershed water and energy balance, diagnose their possible causes, and provides
directions for local correction.

The physical diagnosis makes sure that a GPD represents a plausible water-energy balance in a glacierized catchment. The
water limit helps to identify the additional water term in the water balance or missing water-input. This gives directions to
correct the underestimated GPDs based on the rational water and mass balance in a glacierized catchment. For example,
APHRO, TRMM, and PGMFD are out of water limit ([]), and the missing amount of precipitation in these datasets may be
the undercatch and undetected solid precipitation at higher elevations (Rasmussen et al., 2012). If such datasets are being
used in hydrological simulations, they would result in a highly negative mass balance in the long term and simulate
implausible conditions in the glacierized catchment. These datasets are insufficient to close the realistic water-energy
balance in a glacierized watershed. The simulated hydrology using these datasets is underestimated in glacierized catchments
([d). On the other hand, for example, CFSR and HAR are mostly out of energy limit ([), and the overestimated water-input
will result in higher storage ([), and it may simulate implausible positive mass balance conditions in glacio-hydrological
modeling. However, the higher inter-annual variations in these GPDs ([b) make the inter-annual variations in simulated mass
balance very high ([e). The under- or over-estimated GPDs can be corrected based on the rational output of streamflow and
glacier changes in a glacierized catchment.

The physical diagnosis of GPDs in UIB indicates that GPDs may not reproduce the true water balance, and most of them
might be unsuitable for hydrological applications in such glacierized catchments. Similar concerns were highlighted by
Dahri et al. (2016), who performed an evaluation of GPDs and concluded that these are not suitable to force the hydrological
models in UIB. The runoff peak lags about four months behind the precipitation peak in the Himalaya, Hindukush, and
Karakorum sub-regions ([). The distribution of runoff and precipitation in such a manner highlights the higher solid
precipitation in winter and the dominance of meltwater contributions during the summer season in UIB. Similar arguments





have been made by several researchers in the region, e.g., (Hewitt, 2007; Khan et al., 2015; Lutz et al., 2016; Mukhopadhyay and Khan, 2014). The annual runoff is higher than precipitation for APHRO, TRMM, and PGMFD averagely ([), which highlights that these datasets would cause a negative change in glacier storage as additional meltwater term may be needed

to compensate the water balance in the region, which is the case in UIB (Immerzeel et al., 2015).

The physical diagnosis component in Tri-approach also helps in detecting the possible effects of meltwater on the changes in glacier storage based on the water-energy and mass balance. The problems in GPDs can be diagnosed by analyzing the physical factors involving in these effects. The physical diagnosis identifies the "True", "Gaining", and "Leaky" catchments based on the water-input into the catchment. UIB was identified as "Gaining" catchment based on APHRO, PGMFD, and

TRMM, whereas it was "Leaky" based on CFSR and HAR ([). There are three possible reasons for the case of "Gaining" catchment: (a) additional water contribution from glacier melt (characterized by a negative change in glacier storage in [), which is the case in UIB (Immerzeel et al., 2015), (b) underestimated precipitation (Valéry et al., 2010), which is true for APHRO, PGMFD, and TRMM in UIB ([) and (c) errors in runoff measurements (Andréassian and Perrin, 2012). Underestimation of precipitation and additional water term in water balance is evident from the conclusions of previous

studies in UIB (Dahri et al., 2016; Immerzeel et al., 2015; Lutz et al., 2014). Similarly, Rasmussen et al. (2012) found that the chances of snow undercatch might be as high as 20−50% in high altitude mountainous areas, which is the case in UIB, especially, in the Hindukush and Karakorum sub-regions. The possibilities of runoff measurement errors have been warned in different studies in the region (Mukhopadhyay and Khan, 2014).

On the other hand, for the "Leaky" catchments, there might be four reasons in addition to discharge measurement errors

(underestimation): (a) errors in the estimation of $ET_p$ (underestimation; [), (b) overestimated precipitation, which is the case for CFSR and HAR ([), (c) higher infiltration or local aquifer recharge, or (d) underground water flow towards another aquifer (Andréassian and Perrin, 2012). The impact of inter-catchment groundwater flow on the behavior of "Leaky" catchments has been analyzed in France by Le Moine et al. (2007), who suggest that underground water affects the overall water balance. However, in UIB, the most probable reason for "Leaky" catchment behavior under CFSR and HAR is the

overestimation of precipitation in these datasets ([; [; [). The bed is rocky, and vegetation is very low in UIB, such conditions strengthening the conclusion of overestimated CFSR and HAR precipitation. Several researchers (Blacutt et al., 2015; Liu et al., 2018; Silva et al., 2011) have provided evidence for overestimated CFSR precipitation in different parts of the world.

The actual over- and under-estimations identified based on physical diagnosis provides the basis to kick-off the correction process of GPDs. The violations of water and energy limits by GPDs give an idea of the impacts of meltwater contributions

and glacier mass storage in the catchment, and thus, to adopt the proper correction technique.

### 4.3 Ability to investigate the rational output of simulated hydrology and glacier changes

The practical simulation component in the Tri-approach framework investigates the rationality of simulated hydrology and glacier changes, and it ensures the ability of a GPD to represent the balanced outputs of glacier changes and streamflow simultaneously. When the GPDs are underestimated, for example, APHRO, TRMM, and PGMFD ([; [; [), they are





insufficient to reproduce the hydrology and mass balance at the same time. Adjusting one of them during the calibration
        process would cause underestimations in the other in simulated results ([). When the GPDs are overestimated, for example,
        CFSR and HAR in the case study ([; [; [), they may overestimate at least one of the simulated streamflow or glacier mass
        balance ([), whereas, the opposite is true for the underestimated GPDs. The main reason for such results is that the amount of
        precipitation in these datasets is underestimated as compared to the observed runoff ([), and most of them are unable to
represent the realistic water-energy balance in UIB ([). Hence, the uncertainties in the simulated outputs of glacio-
        hydrological applications can be identified using the Tri-approach framework.

        The quantitative investigation outputs of the Tri-approach framework ([; [) are useful for the correction of GPD and multi-
        parameter calibration during glacio-hydrological simulations. The practical simulation, in combination with the statistical
        and physical diagnosis, helps in investigating the inter- and intra-annual variations based on a rational balance between
hydrology and glacier changes ([). The Tri-approach framework helps to avoid the risk of equifinality as it provides
        directions for the multi-parameter calibration based on the quantitative outputs of diagnosis of water-input and output in a
        glacierized watershed ([). The underestimated precipitation may be compensated with other water balance components, e.g.,
        evapotranspiration, snow, or glacier melt (Ragettli and Pellicciotti, 2012; Schaefli, 2005; Shafeeque et al., 2019). A false
        calibration parameter set would enhance the simulated meltwater to reduce the BIAS between simulated and observed runoff
(Ragettli and Pellicciotti, 2012; Wang et al., 2018). For example, in the case of the SPHY model forced by APHRO,
        PGMFD, and TRMM precipitation datasets ([) to enhance the simulated runoff, a higher negative mass balance would result.
        However, keeping the average mass balance closer to observed mass balance data ([) eliminated the risk of equifinality and
        avoided the glacier compensation effect. At the same time, it was confirmed that combined contributions of snow and glacier
        melt during the spring and summer seasons were higher (69% and 53%) for overestimated GPDs (CFSR and HAR,
respectively) and lower (-44%, -27%, and -35%) for underestimated GPDs (APHRO, PGMFD, and TRMM, respectively) as
        compared to that of OBS ([). If an underestimated precipitation dataset is generating sufficient runoff in a glacio-
        hydrological simulation, then it is for sure that the glaciers are compensating that amount of runoff. Therefore, the simulated
        results in such a situation are questionable.

        Based on the quantitative results of the current study, it is concluded that GPDs generally cannot reproduce the rational
output of glacier changes and hydrology in glacierized catchments. Therefore, it is recommended to correct the GPDs based
        on the local mass and water balance in glacierized catchments before any hydrological application.

### 4.4 Uncertainties

        The application of the Tri-approach in diagnosing the GPD for glacio-hydrological simulations in mountain regions can be
        influenced by uncertainties. A denser observation station network is required, especially at the higher elevations, to reduce
the uncertainties in the observed datasets for hydrological simulations (Li et al., 2020; Liu et al., 2019). This is crucial for the
        southeastern parts of the UIB that are characterized by low station density (Figure 1). Similarly, the observed runoff might
        also be affected due to random and measurement errors. Inconsistencies in the measurements and overlapped measuring





periods for different hydrological stations amplify the overall uncertainty in the data. Besides, the runoff and precipitation have different peak-occurring timings due to multi-sourced precipitation systems (Dahri et al., 2016) with a maximum

proportion of precipitation in the winter and spring seasons (Hewitt, 2007) contrasting to a maximum runoff in the summer season (Mukhopadhyay and Khan, 2014). The availability of glacier data can also be a limitation and cause a certain amount of uncertainty in the Tri-approach results. The limitations associated with the observed datasets are the unavoidable common issue in glacierized regions.

The use of non-dimensional hydrological representation is suitable and advantageous because physical data like runoff,

evapotranspiration, and precipitation are frequently measured or calculated in any watershed (Andréassian and Perrin, 2012). However, the evapotranspiration data are mostly unavailable since more parameters are required to calculate it. The calculated $ET_p$ values are slightly lower (average -11%) than Glabal-ET0 data, which used Penman-Monteith method. It is in line with the conclusions of Zhou et al. (2020), who concluded that $ET_p$ calculated by Hargreaves method is lower than that by Penman-Monteith method. The methods applied to calculate $ET_p$ may affect the overall representation of the hydrological

alternative of the Truc-Budyko plot. It has been argued that the energy limit may also depend to some extent on the chosen $ET_p$ formula (Coron et al., 2015), which may ignore most climatic parameters and use the only temperature data in the current study. This may be the explanation for slightly negative mass balance for CFSR in the Himalaya sub-region and average OBS in UIB ([) even these were in the feasible domain and represent "True" catchments ([). Besides, estimated $ET_a$ ([) might also affect the final changes in glacier storage based on the physical diagnosis.

In the SPHY model, the glaciers are considered as melting surfaces, which cover a grid cell partly or entirely. Moreover, a grid cell can have multiple parts of different glaciers, and it treats them as a single unit within that grid cell. Although the complex glacier processes cannot be resolved explicitly using the SPHY model; however, melting surfaces at a reasonable resolution serves the purpose of this study.

**4.5 Implications of Tri-approach**

The Tri-approach is very useful for the selection of most suitable GPD for glacio-hydrological applications in any glacierized catchment. In glacierized catchments, it is a common understanding that GPDs need correction before any hydrological application. However, a preliminary evaluation of these GPDs is mandatory not only based on climatology but also water-energy and mass balance before any correction. The Tri-approach can provide basic directions for the correction factors based on climatology, plausible water-energy balance, and glacier changes simultaneously, and thus, assist in

adopting the proper local correction of GPDs. Several researchers corrected the GPDs in different glacierized catchments based on climatology (Dahri et al., 2016), conceptual water balance (Khan and Koch, 2018), and vertical gradients and mass balance distribution (Immerzeel et al., 2015; Wortmann et al., 2018). It is important to note that correction may also induce uncertainty in simulated results based on the technique applied. The corrected GPDs can also be verified using the Tri-approach framework. Meanwhile, if there is no option for the correction of GPDs, then one must choose the best data

representing the water-energy and mass balance in glacierized mountain regions. Besides, the Tri-approach detects the key





limitations of GPDs, and thus, helps to identify the expected uncertainties in the outcomes of glacio-hydrological simulations, for example, under- or over-estimations in simulated hydrology, variations in intra- and inter-annual distribution of streamflow and mass balance, deviations from a concurrent rational output of streamflow and glacier change, among the others. The analysis of change in glacier storage is critically important because precipitation patterns can also be influenced

by changes in glaciers (Ren et al., 2020). The understandings developed using the Tri-approach framework are effective for the data generators and algorithm developers to improve their work keeping in mind the application demands for real time scenarios.

**5 Conclusions**

The Tri-approach framework evaluates the GPDs statistically, physically, and practically. Application in the UIB confirms

that it is plausible in the glacierized watershed where rain gauge data are scarce. The approach has the ability to investigate climatology, water-energy balance, change in glacier storage, and rationality of simulated hydrology and glacier changes in mountain glacierized watersheds.

The statistical validation identifies the potential problems in the temporal distribution of the datasets, e.g., APHRO represents the monthly and seasonal distributions and interannual variations (CC > 0.6) but is underestimated. On the other

hand, CFSR and HAR are overestimated and do not represent the inter-annual variations in UIB (CC ≤ 0.3). Reanalysis based GPDs are generally overestimated (77%−106%), whereas, observation and satellite-based GPDs are underestimated (-41% to -24%). The underestimated GPDs would result in an underestimated hydrology when applied in hydrological simulations. The wet and dry duration spells were generally longer for overestimated and underestimated GPDs, respectively. The physical diagnosis based on Truc-Budyko theory identifies that APHRO, TRMM, and PGMFD datasets make the

catchments "Gaining", indicating an additional water term in water balance due to glacier melting, which results in negative glacier storage (-0.42 to -0.34 m w.e. yr$^{-1}$). On the other hand, CFSR and HAR make the catchments "Leaky", highlighting a positive change in glacier storage (0.37 and 0.21 m w.e. yr$^{-1}$, respectively). The actual under- and over-estimation based on physical diagnosis provides the basic directions for local correction of GPDs in glacierized mountain regions. The "Gaining" catchments (characterized with underestimated precipitation) need more input-water (higher precipitation) to sustain water

and mass balance concurrently, whereas, "Leaky" catchments (characterized with overestimated precipitation) need lesser input-water to reproduce the plausible water-energy and mass balance in a glacierized catchment simultaneously.

The glacio-hydrological simulation confirms the findings of statistical and physical diagnosis that GPDs are generally unable to represent the actual water-energy and mass balance in glacierized catchments. The selected GPDs generally cannot fulfill the requirements of the rational output of streamflow and glacier mass balance concurrently in glacierized catchments. It

provides quantitative directions based on under- and over-estimations in simulated streamflow and glacier mass balance for local correction of GPDs for glacio-hydrological simulations in mountain regions.



## 6 Data availability

For hydrometeorological and mass balance data availability, detailed links/references are provided in Table 2. The source code/software of SPHY model, CDO, and DrinC are available at given links: SPHY (https://github.com/WilcoTerink/SPHY); CDO (https://code.mpimet.mpg.de/projects/cdo); DrinC (https://drought-software.com/download/).

## 7 Author contributions

MS and YL jointly developed the concept and methodology of the study. MS performed the computations and the analysis for the case study. MS wrote the first draft of the manuscript. YL edited the current version of the manuscript.

## 8 Competing interests

The authors declare that they have no conflict of interest.

## 9 Acknowledgment

The research work was supported by the Strategic Priority Research Program of Chinese Academy of Sciences (Grant XDA20060301) and National Natural Science Foundation China (NSFC Grant 41761144075). We are grateful to the Water and Power Development Authority (WAPDA) and Pakistan Meteorological Department (PMD) for providing observed meteorological and hydrological data in Upper Indus Basin. The first author is very thankful to the CAS-TWAS President's Fellowship Program (http://www.fellowship.cas.cn/dms/) for providing financial support for his PhD.

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



## 11 Tables

**Table 1. Study area details. Location of hydrological and meteorological stations are depicted in Figure 1. The adjusted precipitation is presented in [.**


| Sub-Region | Area (km$^2$) | Elevation Max (masl) | Elevation Min (masl) | Elevation Mean (masl) | Specific runoff calculation | Runoff (mm/yr) | Met. Stations | Annual adjusted OBS Prcp. (mm/yr) |
|---|---|---|---|---|---|---|---|---|
| Himalaya (HMLA) | 75335 | 8069 | 639 | 4760 | Kharmong/Area + Astore/Area | 688 | Astore, Burzil, Deosai, Rama, Rattu, Qinghe | 598 |
| Hindukush (HNDKSH) | 36445 | 8012 | 317 | 3118 | (Bisham Qila - Bunji + Gilgit-Astore-Kharmong)/Area | 621 | Bunji, Gupis, Shendure, Ushkore, Yasin, Zani Pass | 551 |
| Karakorum (KRKRM) | 61656 | 8569 | 952 | 4738 | (Bunji - Kharmong - Gilgit)/Area | 590 | Gilgit, Hushey, Khunjrab, Naltar, Skardu, Ziarat | 466 |
| Upper Indus Basin (UIB) | 173,435 | 8569 | 317 | 4406 | Bisham Qila | 450 | All | 540 |



**Table 2. Details of data used in the current paper.**

| Name | Description | Agency | Reference/Link |
|---|---|---|---|
| **Observed climate** | Precipitation and Temperature at 18 stations (1951–2015) (Figure 1d; Table S1) | WAPDA, PMD, and CMDSN | http://www.wapda.gov.pk/ http://www.pmd.gov.pk/en/ http://data.cma.cn/ |
| **Hydrology Data** | Discharge at significant stations (1969−2010) (Figure 1e; Table S2) | WAPDA | http://www.wapda.gov.pk/index.php/component/content/article?id=54 |
| **Potential and Actual ET** | Global Reference Evapo-Transpiration (Global-ET0); Esri_hydro "average annual actual evapotranspiration" | CGIAR-CSI GeoPortal; ESRI | Antonio and Zomer (2018); ESRI (2019) |
| **Land use and land cover data** | Globcover land cover maps (Figure S3) | European Space Agency | (Arino *et al.*, 2012) |
| **Soil data** | Harmonized World Soil Database (HWSD V1.2) (Figure S4) | Food and Agriculture Organization (FAO) of the United Nations | http://www.fao.org/soils-portal/soil-survey/soil-maps-and-databases/harmonized-world-soil-database-v12/en// |
| **Precipitation - Highly-Resolved Observational Data Integration Towards Evaluation (APHRODITE; hereafter APHRO)** | ~25 km (~0.25 degree); Asia; Daily; (1951−2007); Interpolated | Meteorological Research Institute Japan | Yatagai et al. (2012) |
| **Climate Forecast System Reanalysis (CFSR)** | ~30 Km (~0.3 Degree); Global; Daily; (1979−2014); Reanalysis | National Center for Environmental Prediction | Saha et al. (2010) |
| **The High Asia Refined analysis (HAR)** | 10 Km, 30 Km; High Asia; Daily; (2001−2013); Downscaled Reanalysis | Technischen Universität (TU) Berlin | Maussion et al. (2014) |
| **Princeton Global Meteorological Forcing Dataset for Land Surface Modeling (PGMFD)** | ~25 km (~0.25 degree); Global; Daily (1961−2016): Reanalysis and Observation | Princeton University | Sheffield et al. (2006) |
| **Tropical Rainfall Measuring Mission (TRMM) 3B42 V.7** | ~25 km (~0.25 degree); Global; Daily (1998−2017); Remote Sensing | National Aeronautics and Space Administration (NASA) | Huffman et al. (2010); Huffman and Bolvin (2015) |
| **Mass balance data** | West Himalaya, East and Central Karakorum, Hindukush (~2003-2009) For selected areas and glaciers | Literature (-0.45±0.13 m w.e. yr$^{-1}$ to +0.11±0.14 m w.e. yr$^{-1}$) Average = -0.173 m w.e. yr$^{-1}$ | Brun et al. (2017); Gardelle et al. (2013); Kaab et al. (2012); Kääb et al. (2015); Muhammad et al. (2019) |





**Table 3. Calibrated parameters for the SPHY model in UIB.**

| Parameter | Description | Initial range | Calibrated value |
|---|---|---|---|
| $DDF_{glacierCI}$ | Degree day factor debris-free glaciers (mm $°C^{-1}$ day$^{-1}$) | 1.5−9.0 | 5.0±0.5 |
| $DDF_{glacierDC}$ | Degree day factor debris-covered glaciers (mm $°C^{-1}$ day$^{-1}$) | 1.5−9.0 | 4.0±0.5 |
| $DDF_{snow}$ | Degree day factor snow (mm $°C^{-1}$ day$^{-1}$) | 1.5−9.0 | 3.5±1.0 |
| $Snow_{SC}$ | Water storage capacity of snowpack (mm mm$^{-1}$) | 0.2−0.8 | 0.5 |
| $α_{GW}$ | Baseflow recession constant | 0.001−1.0 | 0.05 |
| $BF_{days}$ | baseflow recession days (days) | 90−150 | 110 |
| $k_x$ | Routing recession coefficient | 0.5−0.99 | 0.85 |
| $G_{RF}$ | Glacier runoff fraction | 0−1 | 0.80 |
| $T_{threshold}$ | Threshold temperature (°C) | 0-3.0 | 1.5±0.5 |
| $T_{LR}$ | Temperature lapse rate (°C m$^{-1}$) | -0.0098 to -0.0050 | -0.0065 |

**Table 4. PBIAS (%) in annual precipitation of GPDs against OBS over varying periods from 1951-2017 in UIB.**

| Region | HMLA | HNDKSH | KRKRM | UIB |
|---|---|---|---|---|
| APHRO | -69% | 10% | -62% | -40% |
| CFSR | 61% | 146% | 145% | 106% |
| HAR | -2% | 118% | 132% | 77% |
| PGMFD | -38% | 9% | -44% | -24% |
| TRMM | -53% | -2% | -56% | -37% |





## 12 Figures

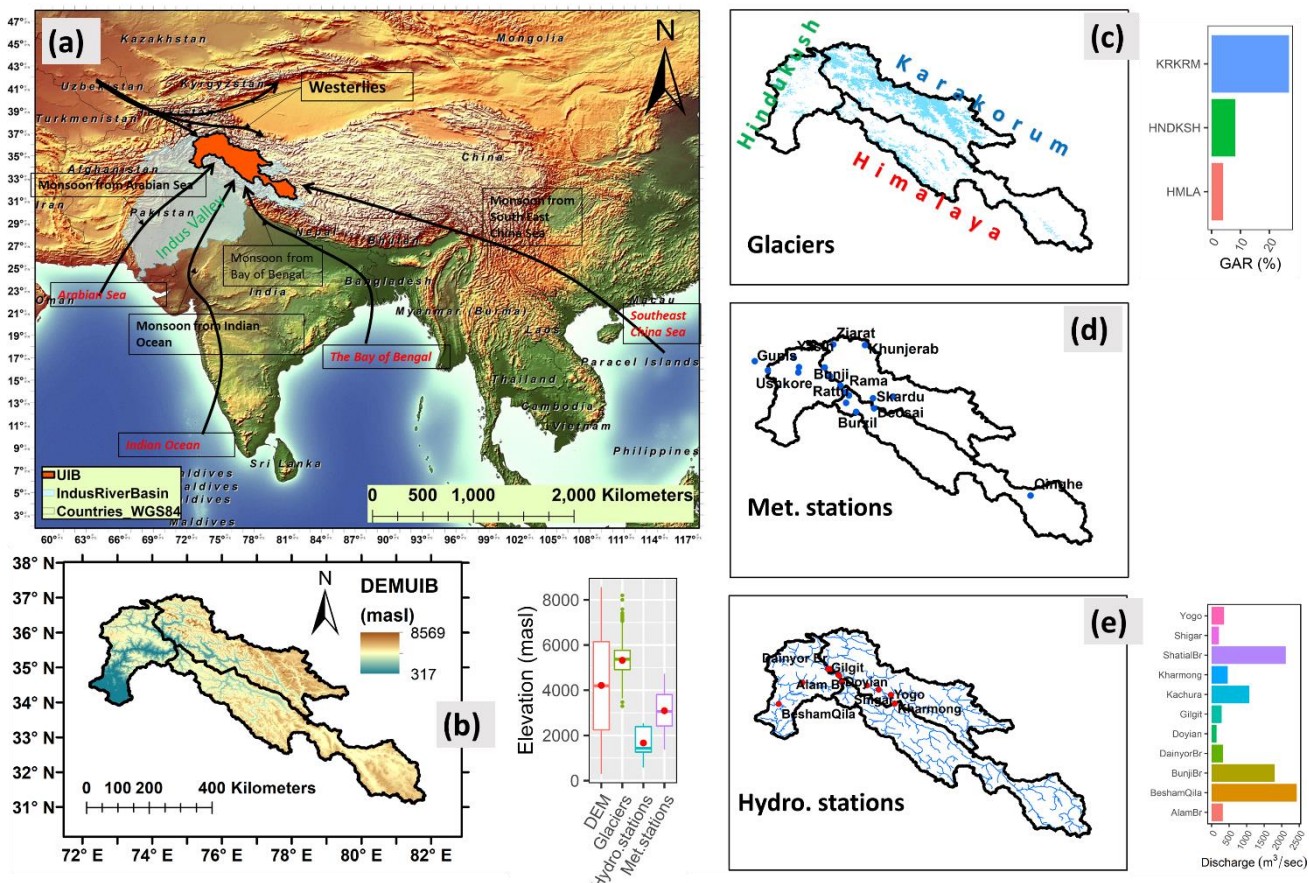

**Figure 1.** (a) Precipitation sources trajectories—westerlies and monsoon. The background layer contains the Countries and Shaded Relief layers from the Natural Earth dataset (http://www.naturalearthdata.com/downloads/). (b) The geographic location of Upper Indus Basin (UIB) and elevation, Digital Elevation Model (DEM) obtained from the U.S. Geological Survey (http://www.usgs.gov), box plots representing the elevations of glaciers, meteorological and hydrological stations, (c) glaciers (Bajracharya and Shrestha, 2011) and glacier area ratio (GAR) in each sub-region (d) location of meteorological stations, and (e) location of hydrological stations and mean discharge at hydrological stations in UIB. Here, KRKRM = Karakorum, HNDKSH = Hindukush, and HMLA = Himalaya.





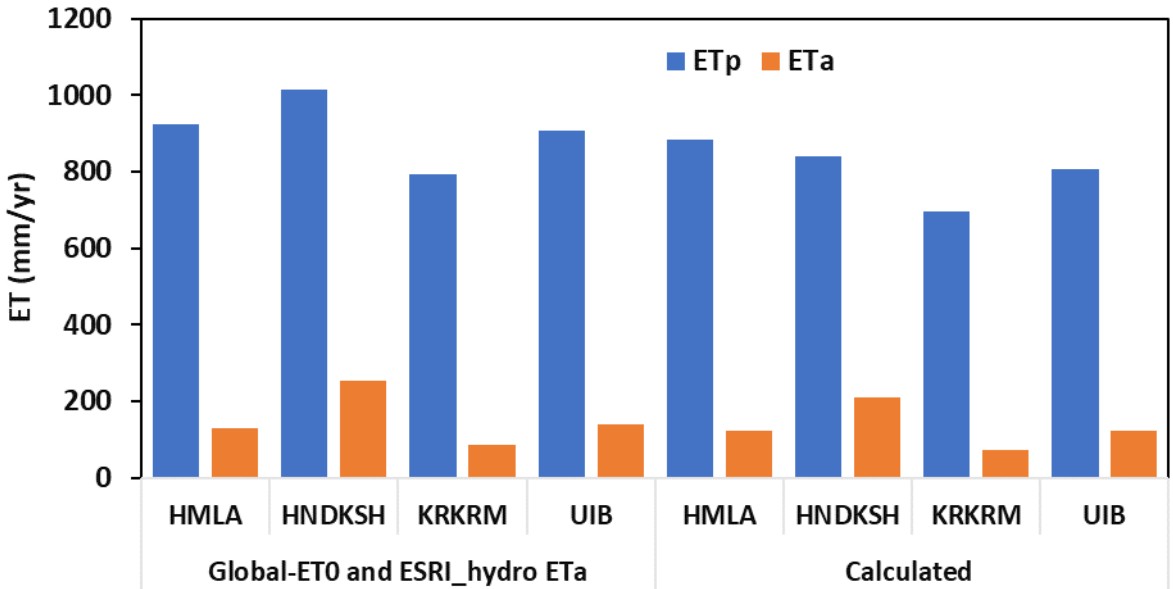

**Figure 2. Values of potential evapotranspiration and actual evapotranspiration. The extracted values are based on the Global Reference Evapo-Transpiration (Global-ET0) and Esri_hydro "average annual actual evapotranspiration". The calculated values are obtained using DrinC (Hargreaves method) and (1.**







**Figure 3. (a) The average annual observed and adjusted precipitation at selected stations. (b) Spatial distribution of adjusted precipitation in UIB.**






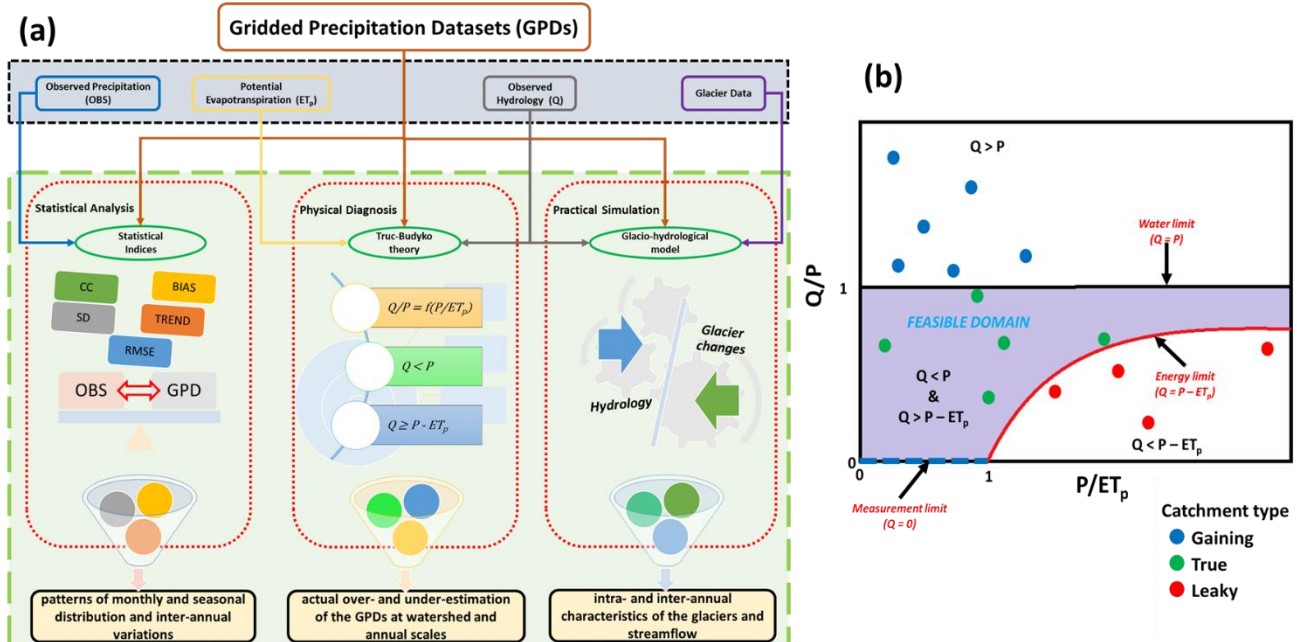

**Figure 4. (a) Schematic diagram of the Tri-Approach framework. (b) Schematic representation of the hydrological alternative of the Truc-Budyko plot. The dots represent catchments. The position of catchments would be different for different precipitation datasets. Here *P* = precipitation; *Q* = Runoff, *ETp* = potential evapotranspiration.**





**Figure 5. Spatial distribution of precipitation based on the selected GPDs in UIB. The temporal distribution of annual precipitation in different regions of UIB from 1951-2017.**


**Figure 6. Graphical representation of performance evaluation of selected GPDs in different regions of UIB at the annual scale.**





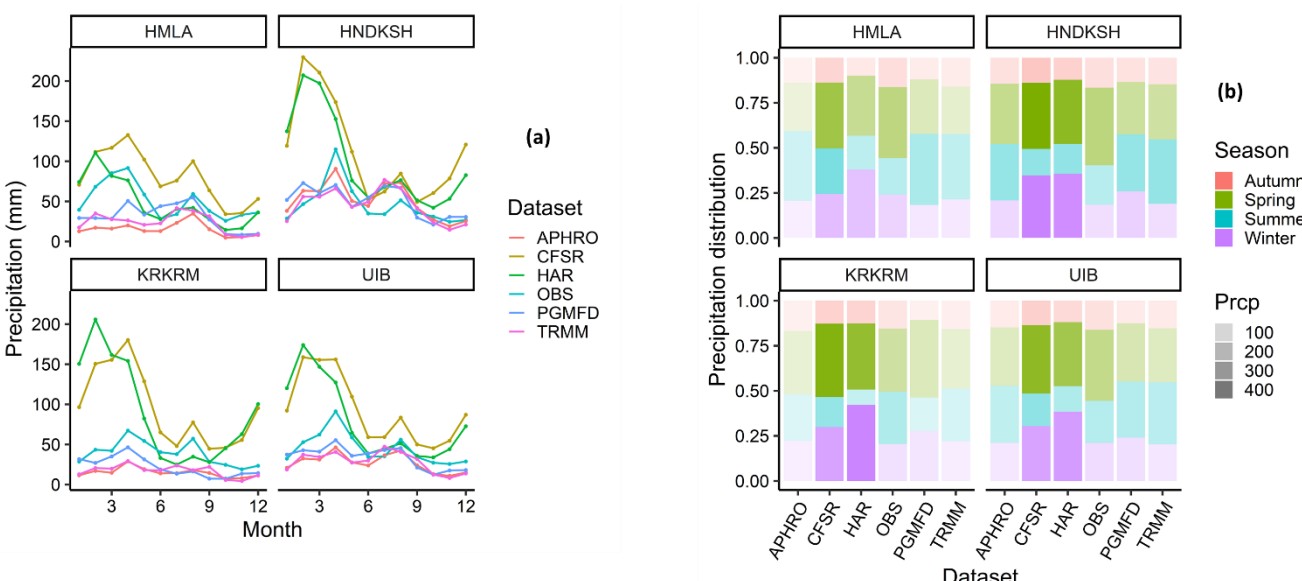

**Figure 7. Distribution of precipitation in sub-regions of UIB at (a) monthly and (b) seasonal scales based on different datasets.**





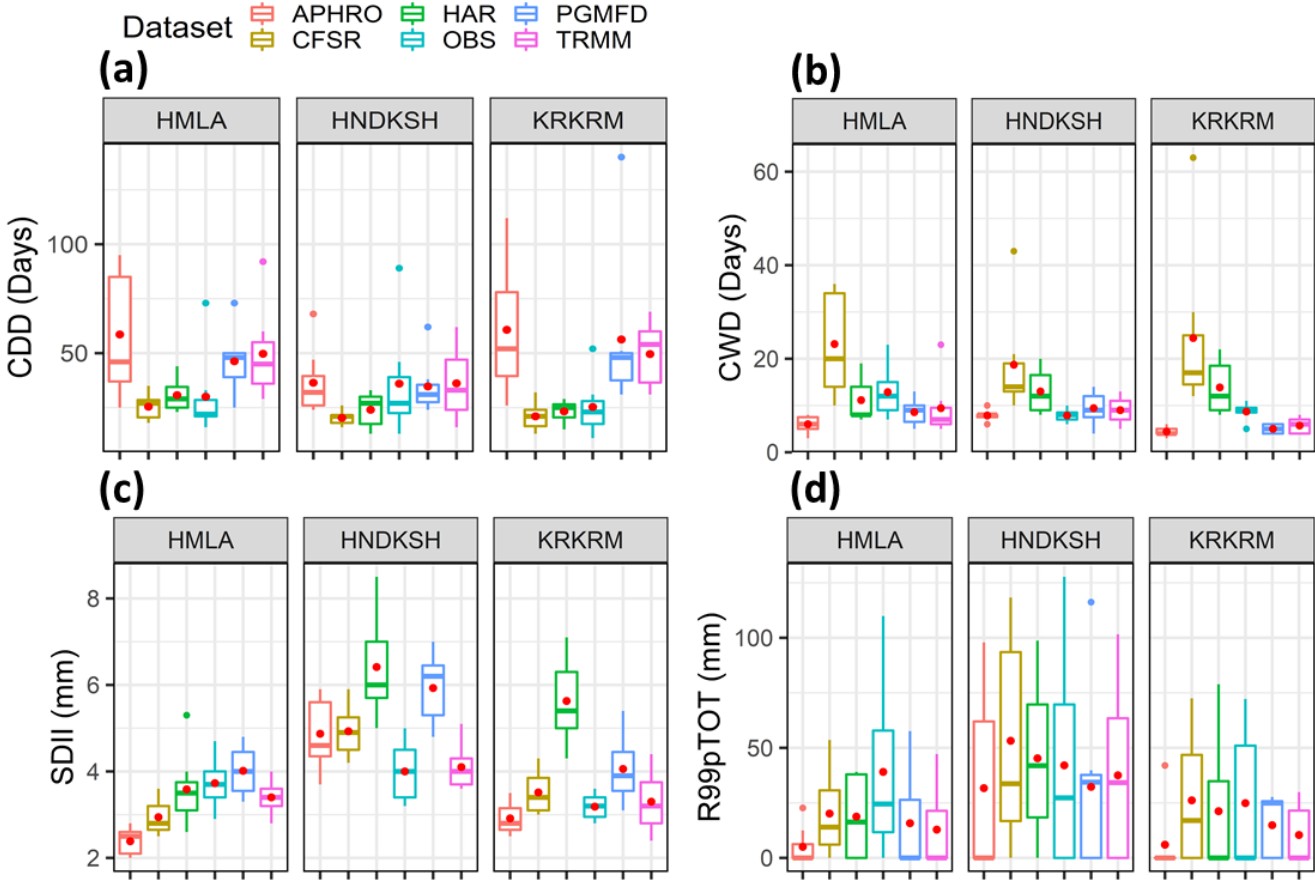

**Figure 8. Selected ETCCDI values for comparing the precipitation extremes in all the datasets over UIB.**





**Figure 9. Comparison between the specific runoff and precipitation (*Q/P* = runoff coefficient) based on OBS and selected GPDs at the monthly scale. The black dashed horizontal line represents the line *Q/P = 1*.**





Figure 10. Comparison between annual specific runoff and precipitation for OBS and selected GPDs in UIB.



**Figure 11. Truc-Budyko plot representing the water-energy balance in UIB.**





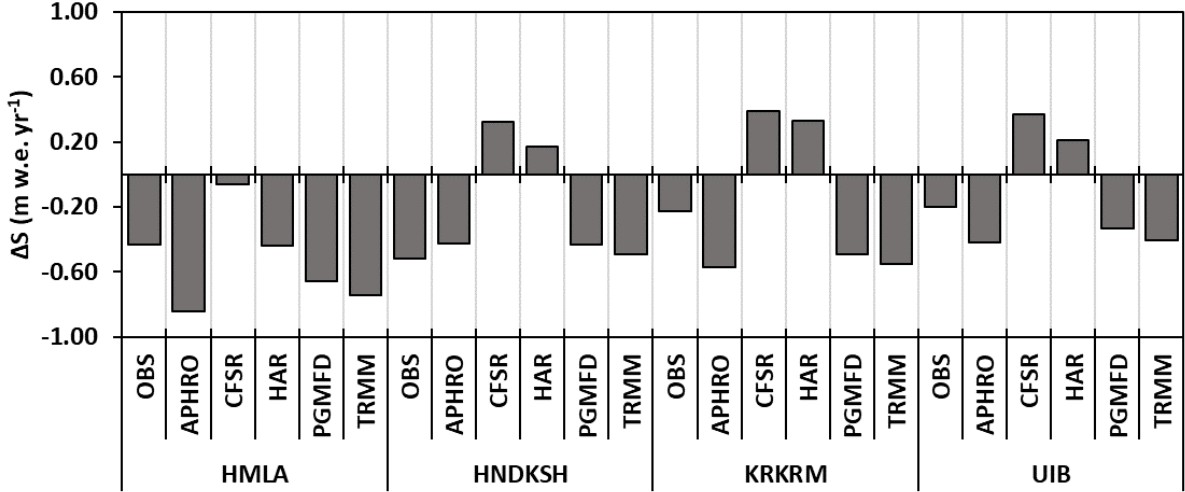

**Figure 12. Glacier storage change based on the water-energy and mass balance in different sub-regions of UIB.**



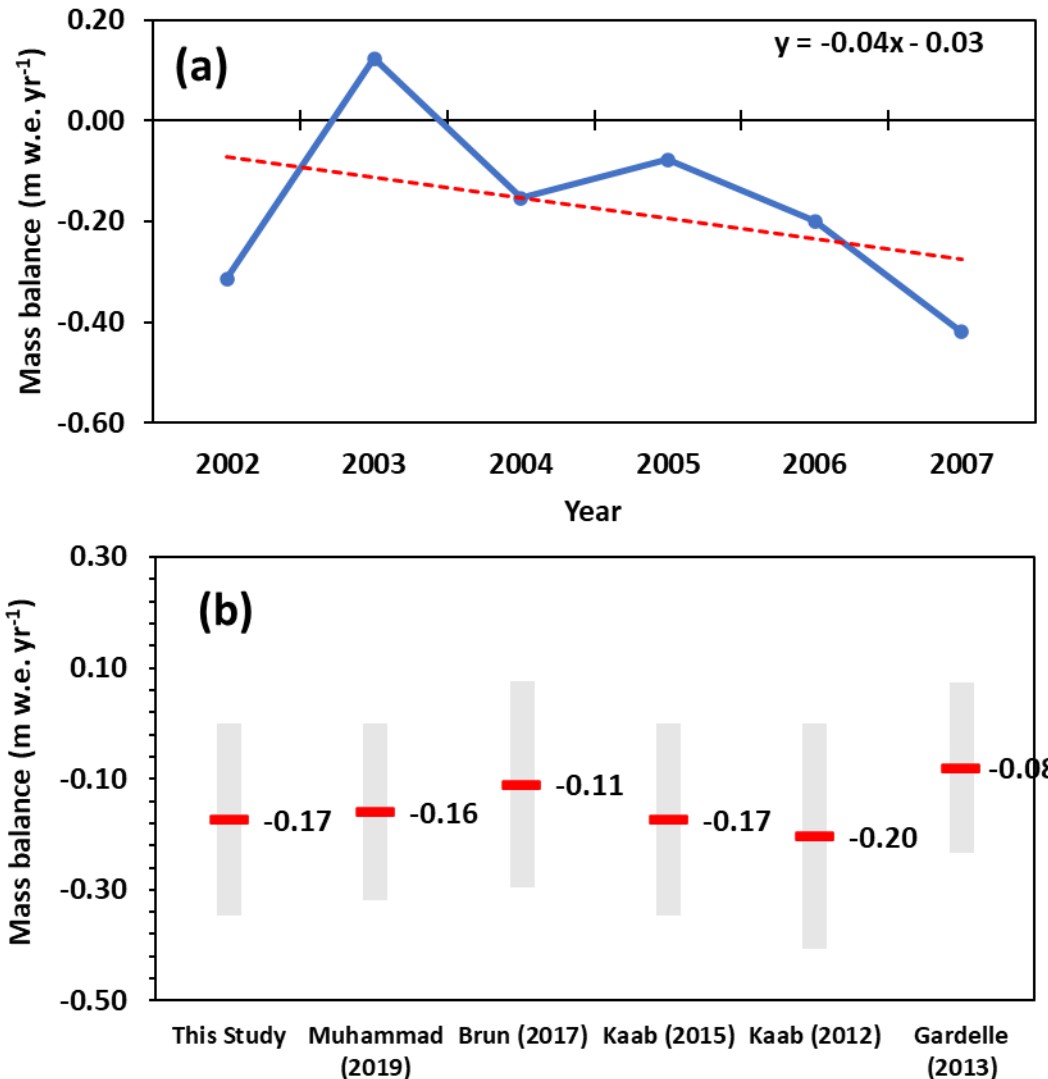


**Figure 13. Calibration of SPHY model based on glacier mass balance. (a) Simulated mass balance based on the calibrated SPHY model in UIB for 2002-2007 and (b) comparison of simulated glacier mass balance against the literature-based glacier mass balance in UIB (~2003-2009).**





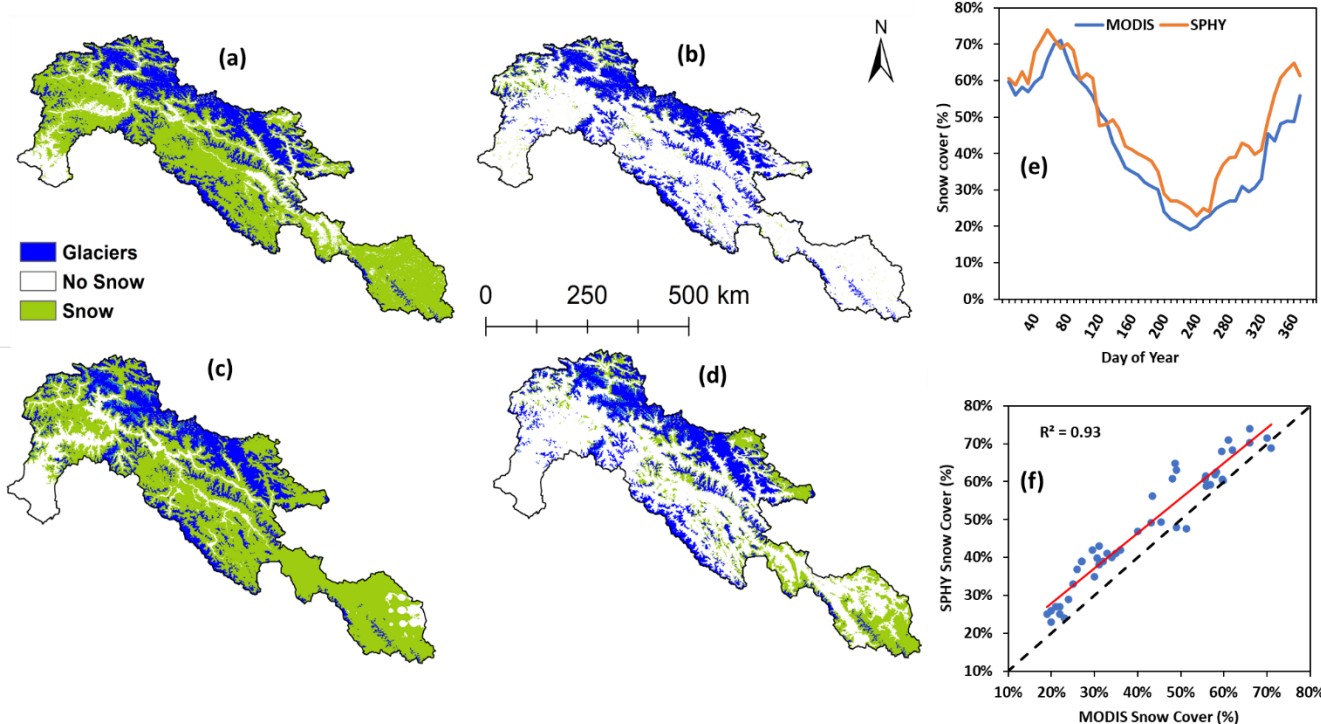

**Figure 14. Calibration of SPHY model based on the snow cover data. (a & c) Maximum snow cover extent in March based on MODIS and SPHY, respectively. (b & d) Maximum snow cover extent in August based on MODIS and SPHY, respectively. (e-f) Comparison between the snow cover based on MODIS and SPHY at an 8-day time scale.**

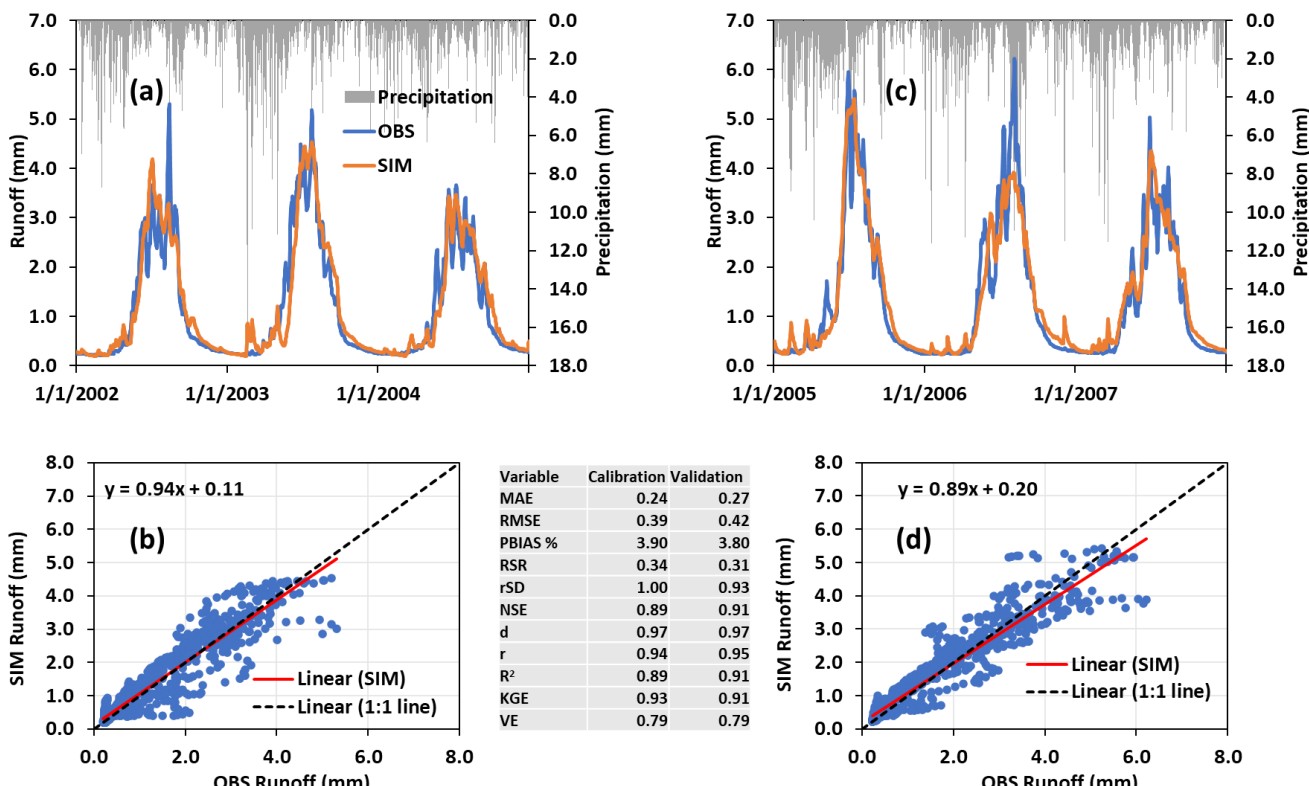

**Figure 15. Calibration of SPHY model based on specific runoff at daily time scale in UIB. (a-c) Calibration results for simulated runoff versus observed runoff (2002-2004). (c-d) Validation results for simulated runoff versus observed runoff (2005-2007). The table shows the goodness of fit results.**


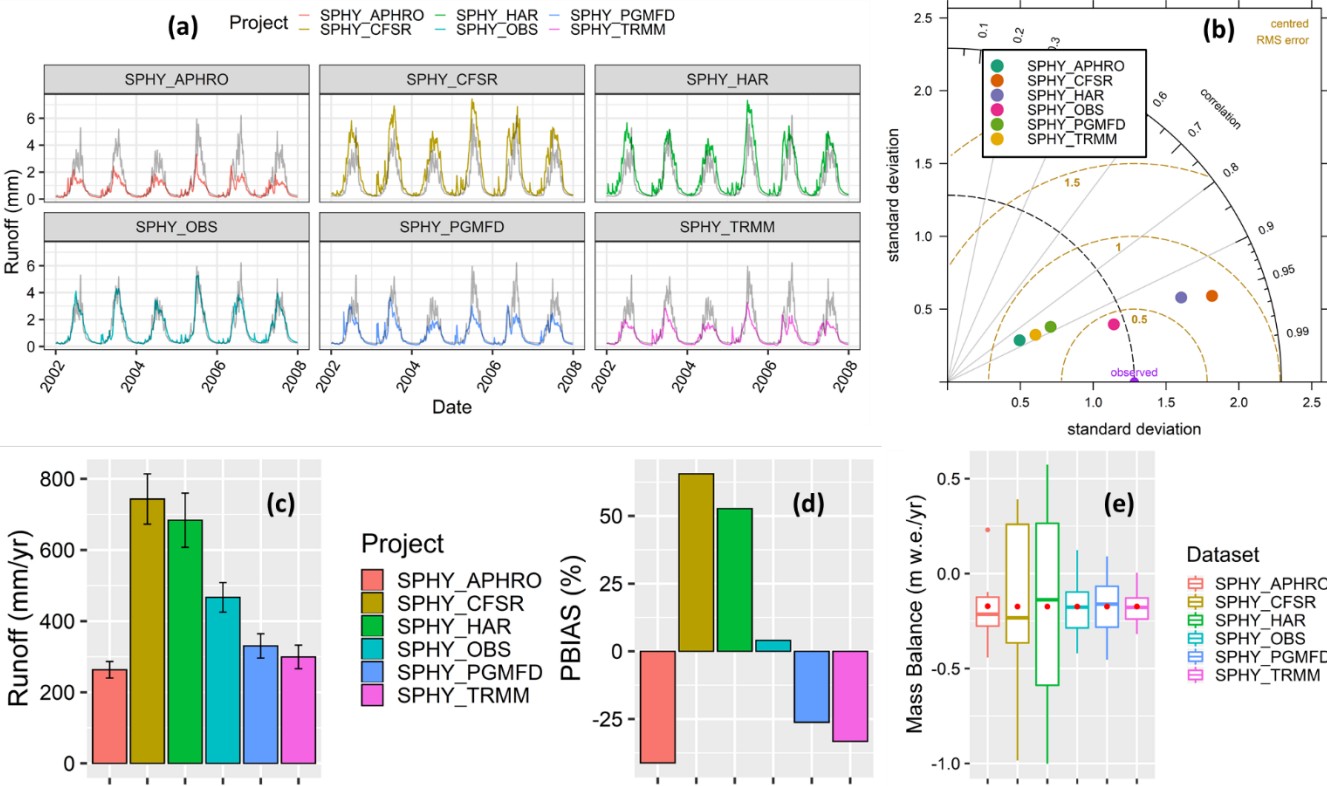

**Figure 16.** **(a) Simulated daily runoff based on selected GPDs for 2002-2007 in UIB. (b) Taylor's diagram to represent the performance of SPHY projects to simulate daily runoff. (c) Average annual simulated runoff. (d) PBIAS for a simulated annual runoff against the observed runoff in UIB. (e) Simulated glacier mass balance for SPHY projects based on different precipitation datasets.**








**Figure 17. (a) Monthly distributions of runoff components for different SPHY projects in UIB. The Pi-Charts represent the annual contributions of runoff components to total runoff. (b) Distribution of runoff components at the seasonal scale.**