# Peer review of "A Tri-Approach for Diagnosing Gridded Precipitation Datasets for Watershed Glacio-Hydrological Simulation in Mountain Regions"

_Hydrology and Earth System Sciences, 2020_

## Author Comment (AC1) · 10 Aug 2020

Dear Anonymous Referee RC1,

Thank you for the assessments and valuable comments. The comments and queries are really helpful to sharpen the revised version of our manuscript. We will try to incorporate the suggestions in the revised manuscript after receiving comments from other referees as well as the editor. The detailed point by point response is given below.

Interactive Response to the Comments of Anonymous Referee #1 General

[Figure]

Comment 1:

The paper presents a model study on the Indus basin, in which different precipitation reanalysis products and observations are used to classify the hydrological behavior of the basin. The precipitation products used are either reanalysis or interpolated gridded observation data. The authors propose a tri-faceted approach consisting of statistical analysis, physical diagnosis, and practical simulation.

Response:

Thank you for the assessments and quick insight of the study framework.

Comment 2:

The study constitutes a contribution toward the modeling of effects of precipitation products on the simulated water balance of the Upper Indus basin, a major source of water for the Indus irrigation system. Inferences are made on the ice mass balance, a topic of high relevance for climate impact analysis in this strongly glacierized system. Depending on products used the UIB turns out to be either a gaining or a losing glacial mass system. However, the approach does not provide any clues on how to assess, which one of the reproduced behavior is closer to reality. As such it provides mainly suggestions on how to correct individual GPDs to match observed outflows, while making unverifiable inferences on ice mass accumulation/depletion.

Response:

Thank you for your comments and query. Glaciers complicate the hydrological processes. Meanwhile, the glaciers are mostly located in the high elevation zone of the basins. What is most challenging for the basin-wide hydrological simulation is that ground truth data of precipitation is commonly unavailable in the high elevation zone. Instead, GPDs derived from different sources by the different techniques are widely used in the simulation. As indicated in other comments, the GPDs may be error-effected, and thus affect the reliability of the simulation. This work is focused on an

integrated framework for evaluating the GPDs and finding out the direction of the local corrections to the GPDs. The UIB has a relatively high ratio of glacier coverage and has been taken as a benchmark for the application of the proposed evaluation framework, demonstrating how the framework is applied to a set of GPDs and some findings. In reality, the glacio-hydrological behavior of the UIB is clearly 'gaining', which is evident based on the negative mass balance and meltwater contributions to streamflow (Bolch et al., 2019; Gardelle et al., 2013; Immerzeel et al., 2015; Muhammad et al., 2019). Here, the proposed approach efficiently captures the behaviors under different products, i.e., 'gaining' for APHRO, TRMM, and PGMFD, whereas, 'leaky' for HAR and CFSR (Figure 11). All the produced behaviors are compared with the reality "actual conditions" in the basin. The extent of 'gaining' behavior based on APHRO, TRMM, and PGMFD is quantitatively different from the reality, as presented in Figure 12. Therefore, the actual situation is not represented by any of these products. Based on the water-energy and mass balance calculations, it turns out that the products responsible for 'gaining' state (APHRO, TRMM, and PGMFD) are underestimated, and the products responsible for 'leaky' state (HAR and CFSR) are overestimated compared to the real or true water balance in UIB (can also be seen in Figure 10-12). Therefore, we recommended a local correction of GPDs before their applications in glacio-hydrological models. The corrections for the under- and over-estimated GPD may be an addition or subtraction of a quantity of water to make them represent the real situation of water balance in the basin. The actual over-and under-estimation can be different at different spatiotemporal scales in different basins (or sub-basins). We provide the quantitative estimations of actual under- and over-estimation in terms of change of glacier storage (Figure 12). As we assumed, groundwater and surface water storage negligible because baseflow/ return flow from shallow groundwater compensates this term in the water balance (Alley et al., 2002; Andermann et al., 2012; Savoskul and Smakhtin, 2013), so all the underestimation is considered as glacier mass loss, whereas all the overestimation as glacier mass gain. The correction of GPDs is suggested to get the rational outputs of both glacier changes (e.g., mass

balance) and hydrology (e.g., streamflow) outputs, concurrently. The simulation results were obtained after an absolute (multi-objective function) calibration and compared scientifically [line: 299-314]. We clearly presented the calibration (Figure 13, Figure 14, Figure 15) and quantitative comparison results (Figure 16, Figure 17), which support our inferences on glacier mass balance. Introduction

Comment 3:

The introduction and other parts of the paper do not mention previous published work by Reggiani et al. (2017) and Reggiani and Rientjes (2015) which is concerned with the basin and compares precipitation reanalysis products for the area of interest. The study also contains a basic water balance analysis for the Indus used to infer on the ice mass balance by specifying the individual terms of the mass balance equation. Reggiani et al 2016 provide an uncertainty analysis for the Shigar subbasin based on a Bayesian analysis of multiple precipitation products.

Response:

Thank you for the recommendations. The suggested studies may be useful to support our research work, so we will try to include them in our introduction and discussion sections, while focusing on the scope of our hypothesis and objectives. However, according to our understanding, Reggiani and Rientjes (2015) and Reggiani et al. (2017) did not specify the individual terms of mass balance. They applied a simple water balance equation and made inferences on glacier mass balance based on the variations in river flows.

General comments

Comment 4:

Overall, it is not clearly explained, what the authors want to demonstrate. Different precipitation products can show very different precipitation depths, while temperatures are generally more consistent among products. For example, Reggiani and Rientjes

(2015) have already shown inconsistencies between reanalysis data and the TRMM as well as CRU data, whereby the latter two heavily underestimate precipitation, leading to different water balance results and conclusions of the ice mass balance when applied to the UIB.

Response:

Thank you. Yes, different GPDs have different precipitation depths, which will lead to different hydrological behaviors, especially when applied in the simulation of a glacierized basin. On one hand, the glaciers complicate the hydrological processes; on the other hand, the information on glacier changes also provides a clue for the GPD evaluation. Based on these considerations, this work proposed a framework for GPD evaluation in a glacierized basin, which is to find out how a GPD represent the reality of a glacierized basin, which GPD performs better than others when GPDs are compared to each other, and the directions to locally correct the GPD. We need to sharpen the objective definition in the introduction section [line: 104-105, 109-112] in revision. As an application of the proposed tri-approach in UIB, we estimated this actual under- or over-estimation for all included datasets (Figure 12), and highlighted that some GPDs need more water to represent the plausible water and mass balance in the basin, whereas, some are just overestimated (Figure 16). The use of over- or underestimated datasets in glacio-hydrological modeling would lead to implausible results and conclusions. Besides, an important intention is to keep the approach and its application (the evaluation/correction analysis of GPDs) as simple as possible because it should be taken as a supplemental primary step rather than the main experiment/modeling in a research study. Yes, you are right; the temperature is more consistent among the different datasets and has no significant difference with the observed data. Therefore, we did not include the temperature analysis in our study. The physical diagnosis characterizes the glacio-hydrological behavior of a basin under different GPDs (Figure 11). It also provides the actual under- or over-estimation of precipitation in different GPDs in terms of glacier storage (Figure 12). The practical simulation also presents the clear

differences among the simulated hydrology forced by different GPDs. The rationality of the glacier and hydrological outputs (concurrently) are investigated (Figure 16 and Figure 17). The results highlighted the need for local correction of GPD before their application in the glacio-hydrological models.

Comment 5:

The use of a tri-approach as proposed here does not give more insights than just using one of the three.

Response:

The Tri-Approach consists of three components, statistical, theoretical, and practical. These three components are step-wise forward. The implementation gets more complicated from statistical to theoretical and practical components. However, more and more insights can be revealed as they get more and more complicated. The researcher can apply this approach just as an initial step instead of a comprehensive analysis before actual glacio-hydrological modeling. This would save their time and efforts to select the datasets and adopt the right local correction in a simple and reproducible way. The approach efficiently can sort out the GPDs to represent the glacio-hydrological behavior of a basin and label them for which state they represent (i.e., 'gaining', 'leaky', or 'feasible'). Moreover, the approach also identifies the actual under- or over-estimation in the selected/included GPDs based on water-energy and mass balance of a basin (Figure 11 and Figure 12). Besides, the practical simulation provides a wider insight into the hydrological behavior of a glacierized basin. As an example, this work used the SPHY model for the practical analysis. The SPHY model projects were tuned to represent the actual glacier mass balance state in the UIB (Figure 13−15). This was done (a) to avoid the equifinality (glacier compensation effect), (b) to ensure the reasonable comparisons among the simulated hydrology (Figure 16−17). Keeping the mass balance closer to the reality during simulation, is the best option to explore the rational outputs of glaciers and hydrology, concurrently, for different precipitation forcings. The observed discharge data is more reliable than the available mass balance for comparison purposes. For such conditions, it is reasonable to use mass balance data for tuning the glacio-hydrological model and the hydrology data for comparisons. The case study in UIB demonstrated how the Tri-approach can help to select a proper GPD and how to correct it locally to represent the "reality". Meanwhile, we are still open to accepting any suggestions to improve the presentation of analysis, which may be helpful to present the work from a different perspective.

Comment 6:

Spatial precipitation products are information, that is inherently error-affected. To draw different conclusions on the water balance using individual products is an interesting exercise, but the ultimate scope unclear. What is the goal of finding out that using one product the catchment turns out to be "leaky" (i.e. it stores water through glacier mass increase) and with the other product the catchment becomes "gaining" (i.e. releases water by glacial melt) and then concluding that corrections to GPDs need to be made accordingly? This does not give any insights into what is actually happening within the basin.

Response:

Thank you. Precipitation is the input of the glacio-hydrological system in a basin. Correct precipitation information is the prerequisite of proper simulation of the basin process. Due to the unavailability of the ground truth data of precipitation in the high elevation zone, the derived GPDs are generally error-affected, as you commented. Diagnosis of the GPDs is, thus, the first thing to do when they are used. The Tri-approach proposed in this work is focused on the diagnosis. Applied to the UIB, the Tri-approach selected the "best" GPDs among five datasets and found out the directions for local corrections. As demonstrated, the proposed Tri-approach provides the simplest way for a quick glimpse of the glacio-hydrological behavior represented by a GPD.

Comment 7:

In my view, the real advantage of having multiple products is their mutual combination and exploitation of informative content as a "package". This point has not been addressed at all by the authors.

Response:

Agree, and thanks for the suggestion. The mutual combination of different GPDs is an effective way to use multiple datasets as an ensembled package for further applications. However, this is not included in the goals of our study. We may consider this suggestion in our future work to develop a direct way of combining different datasets based on their performances to represent the hydrological behavior of glacierized basins. However, this study will mainly describe the shortlisting of the best GPD and prescribe directions for its local correction (if required) based on the actual glacio-hydrological conditions in a basin.

Comment 8:

The products or the hydrological signals derived using these products must be conditioned on available ground information (flows, precipitation, snow area extent etc.), and on this basis a selection made about which products is superior, by attributing it more importance vs other, less informative ones. If done properly, the conditioning should remove bias and reduce the uncertainty given the ensemble of products.

Response:

Agree. In fact, the rationality of glacio-hydrological outputs is conditioned based on the observed flows, precipitation, ET, snow cover, and glacier mass balance in the current study. The adjusted observed precipitation (Figure 3), MODIS snow cover, and observed mass balance are used to parameterize the base model. The model is calibrated for observed mass balance (Figure 13), snow cover (Figure 14), and streamflow (Figure 15). Multi-parameter calibration is adopted to avoid the risk of equifinality (caused by glacier compensation effect). Finally, the SPHY model projects (Figure

16) (one for one precipitation dataset) were tuned for actual mass balance, and then the simulated hydrology outputs were compared. The streamflow data is more reliable and readily available as compared to mass balance data. Therefore model was parameterized with observed mass balance data (Figure 16d), and the results were compared with observed streamflow data (Figure 16c). The calculated ET data may contain some extent of uncertainty; however, we also compared and calculated ET with the authentic derived products (Figure 2). This highlights that the methods adopted in the Tri-approach are properly conditioned based on the observed flows, precipitation, snow cover, and glacier mass balance, which remove bias, reduce the uncertainty, and support the reliability of inferences drawn in the current study.

Comment 9:

I personally see the present study as a collection of GPD applications, that lead to a qualitative classification of GPD products, but do give improved insights into glacio-hydrological behavior or clues on an improved structure of spatial hydrological model forcing.

Response:

Thanks. This work is to propose a framework for quantitative assessments of GPDs, which takes the the observed local information of climatology, hydrology, and glaciology into account. A framework for diagnosing the GPDs based on the ground truth data and theoretical principles is considered a demand of the hydrology community. And this is the reason that we presented our work here. This work thus proposed a Tri-approach framework, which diagnoses the potential problems in the GPDs from multiple perspectives and provides clues about the different data structure of GPDs and their local correction, which eventually helps to improve the glacio-hydrological modeling study and to infer plausible conclusions. It should be pointed out that the Tri-approach stood upon the shoulders of previous researchers, cited (among others) as in the reference list of the main manuscript.

We will revise the manuscript following the valuable comments, hoping that the integrated framework for the GPD diagnosis can be shared and improved through the interested readers.

Sincerely,

Yi Luo and Muhammad Shafeeque

References

Alley, W.M., Healy, R.W., LaBaugh, J.W., Reilly, T.E., 2002. Flow and storage in groundwater systems. Science, 296(5575): 1985-90. DOI:10.1126/science.1067123 Andermann, C. et al., 2012. Impact of transient groundwater storage on the discharge of Himalayan rivers. Nature Geoscience, 5(2): 127-132. DOI:10.1038/ngeo1356 Bolch, T. et al., 2019. Status and Change of the Cryosphere in the Extended Hindu Kush Himalaya Region, The Hindu Kush Himalaya Assessment, pp. 209-255. DOI:10.1007/978-3-319-92288-1_7 Gardelle, J., Berthier, E., Arnaud, Y., Kääb, A., 2013. Region-wide glacier mass balances over the Pamir- Karakoram-Himalaya during 1999–2011. The Cryosphere, 7(4): 1263-1286. DOI:10.5194/tc-7-1263-2013 Immerzeel, W.W., Wanders, N., Lutz, A.F., Shea, J.M., Bierkens, M.F.P., 2015. Reconciling high-altitude precipitation in the upper Indus basin with glacier mass balances and runoff. Hydrology and Earth System Sciences, 19(11): 4673-4687. DOI:10.5194/hess-19-4673-2015 Muhammad, S., Tian, L., Khan, A., 2019. Early twenty-first century glacier mass losses in the Indus Basin constrained by density assumptions. Journal of Hydrology, 574(March): 467-475. DOI:10.1016/j.jhydrol.2019.04.057 Reggiani, P., Mukhopadhyay, B., Rientjes, T.H.M., Khan, A., 2017. A joint analysis of river runoff and meteorological forcing in the Karakoram, upper Indus Basin. Hydrological Processes, 31(2): 409- 430. DOI:10.1002/hyp.11038 Reggiani, P., Rientjes, T.H.M., 2015. A reflection on the long-term water balance of the Upper Indus Basin. Hydrology Research, 46(3): 446-446. DOI:10.2166/nh.2014.060 Savoskul, O.S., Smakhtin, V., 2013. Glacier systems and

seasonal snow cover in six major Asian river basins: hydrological role under changing climate, 150. IWMI.

---

## Short Comment (SC1) · 31 Aug 2020

The manuscript presents a good insight about the runoff generation from UIB using different gridded precipitation products and characterized the basin as leaky and gaining. Adjusted precipitation graph and the map indicates that the precipitation stations at the lower elevations like Gilgit, Bunji and Skardu receives very less amount of precipitation which is established fact thus adjustment factor is maximum for these stations. However, it is mentioned in the WAPDA reports and papers that the maximum precipitation occurs in the elevation zone of approx. 4500 – 5100 m.a.sl. Moreover, as reported

by different researchers and also mentioned in this manuscript that the precipitation is largely affected by topography. Therefore, it would better to consider the topography, which actually plays role in vertical distribution of precipitation, to adjust the precipitation rather than the correction of low elevation precipitations gauges data. One significant information one can reveal from the simulated results of the runoff model that the good correlation with observed flows could be attributed to the increased precipitation (adjusted) of the stations installed at lower elevations. Therefore, increase in precipitation could be further attributed to the excessive melt of glaciers in the ablation zone which corroborate the findings of the previous researches that have reported thinning of glaciers (losing mass) in ablations zones. Authors have well documented this fact and cited relevant literature. Authors have made comparison between the different gridded precipitation datasets and the measured precipitation which provides the interesting insight e.g., HAR dataset underestimates the precipitation in Himalaya and over predicts in the Hindukush and Karakoram (Figure 5). It is noteworthy to mention that Himalaya region is influenced by Monsoon Precipitation and the other two regions are nourished by westerlies. In my opinion, appropriate corrections in HAR dataset may result better runoff simulation which could be considered for future researches. Therefore, I would suggest authors to discuss this aspect to provide the directions for future research. Moreover, HAR data set pattern is in-phase with the observed one and shows the maximum precipitation in winter season for Karakoram and Himalaya regions which also corroborates the previous findings. In addition, I would suggest authors to mention the suitability of different gridded sets for the sub-catchments of UIB for the selection of suitable one based on influential precipitation system. The paper is good contribution to the existing knowledge and can be used as preliminary step before any complex modelling approach using gridded datasets. The directions for the correction of under or overestimated precipitation based on water and mass balance are well explained.
* * *
194, 2020.

---

## Referee Comment (RC2) · Anonymous Referee #2 · 15 Oct 2020

**General comments**

This paper presents a threefold approach to assessing gridded precipitation datasets in mountainous regions. The approach is tested over the Upper Indus Basin, using five precipitation datasets, and observations from precipitation gauges, runoff measurements and glacier mass balance measurements. The first approach is a comparison with observations, the second a balance between precipitation, runoff and evapotranspiration, and the third uses a hydrological model (including glacier and snow components) to compare to observed runoff. Overall, the manuscript presents an interesting approach to resolving the issues of assessing precipitation datasets in data-sparse regions with complex topography. There is also an interesting analysis of the performance of these datasets over different regions of the Upper Indus Basin. In particular, the seasonal analysis over the datasets demonstrates that while most datasets represent the summer monsoon well, there is far more variation over the representation of the winter westerlies.

**Major comment**

My main comment concerns the second approach, referred to as the 'physical diagnosis' in the manuscript, which aims to determine whether a basin can be considered plausibly realistic using a water energy balance. Using the precipitation observations, the authors show that this catchment is not 'physically realistic' as defined using the Truc-Budyko plot and suggest that this is likely to be due to glacier melt and storage. This nicely motivates the use of SPHY, which includes a glacier component, but suggests that this second method is not suitable for assessing precipitation datasets in regions with snow and glaciers, as it is missing an essential physical component. The authors use this to assess glacier change based on each dataset, but it is not clear how this gives extra information about the reliability of the precipitation datasets themselves. In my opinion, this approach could be removed completely (or significantly reduced to just looking at observations and demonstrating why this method is not suitable). This would also help to shorten the manuscript somewhat, as it is currently quite long.

**Minor suggestions**

Some minor suggestions are listed below (I should note that I was not able to see any of the figure or table references in the manuscript, which may have led to some misunderstandings on my part):

Given that not all readers will necessarily be familiar with the statistical plots presented here, it would be useful to have more information in the figure captions, and to define the acronyms used in the figures. E.g. for figure 8 "the boxes represent ….." e.g. for figure 10 "each dot represents …. (an average/total for one year of data?)".

Throughout the manuscript, the authors should check that acronyms and abbreviations are defined where they first appear, and consider repeating these, or providing a nomenclature. For example on line 95, Q, P, ET_p all need defining, on line 210 ETCCDI needs defining.

Line 174: Was this adjusted using the same wind data as used in Dahri 2018? I think given the importance of this adjustment, an extra line explaining it would be useful.

Line 300: Please add references to the sentence 'The observed mass balance data were extracted from the literature'.

Line 305: Where is the data from Besham Qila held or is there a reference for this data?

Line 323/Figure 5: Could you plot the points of the stations onto the map in figure 5 so it's easier to compare the observations and colour maps? More importantly, please switch the colour scheme for either figure 3 or

5 so that they are the same, with the same scale (a divergent colour scheme for both would make it clearer which areas are high precipitation and which are low precipitation). It's tricky to compare them when blue is dry regions in one plot and wet regions in the other.

Line 326: While it's clear what you mean here, I think technically this should be 'the GDPs did not show statistically significant trends', as you cannot generally use statistical tests to prove a lack of trend.

Line 331-333: Are these discussions about bias coming from the Taylor diagram in figure 6? It might be better to talk about RMSE, as that's what you have shown here in figure 6.

Line 333: word missing '….as the better in UIB…' -> '…as the better model in the UIB…'

Line 345: It would be good to emphasize that this under/overestimation is particularly true during the winter, as it's interesting that these datasets appear to represent the summer monsoon much more effectively than the winter westerlies.

Lines 349-355: You could consider cutting figure 7 b and the accompanying text, as I think this is all shown in figure 7 a and that discussion. If figure 7 b is kept, could the seasons be put in order? I'd recommend Winter, spring, summer, autumn as this will make it easier to see the winter and spring precipitation together.

Lines 357-368: It's not quite clear how these numbers relate to figure 8, or how the values in figure 8 and the numbers in the text are calculated. Are the mean and standard deviations given in the text taken from each year? I.e. the maximum CDD taken from a year, and then averaged over all the years? Given figure 8 shows the median and 25$^{th}$/75$^{th}$ percentiles, it might be more useful to discuss those? (although presumably the red dots are the means in each case).

Figure 9/lines 372-382: is the runoff value the same for each of the models? Is this a measured value? Please state in the text and the figure caption.

Lines 383-391: This section discusses correlation between runoff and precipitation. However given there is no significant correlation between the observed precipitation and runoff, except in the Karakoram, it seems that there may be other factors that need to be taken into consideration, and therefore that correlation between these two variables probably should not be used to judge the datasets?

Line 456: Are these six SPHY model runs identical, except for the precipitation dataset used? Or are there some differences, apart from the precipitation datasets?

Line 494-497: I don't quite understand this.

Line 510: Could you add some references here?

Line 589-594: you suggest undercatch in the observations here as a reason for the unbalanced water balance, but haven't you already corrected for this?

Lines 595-598: is increasing glacier mass balance also a reason for a 'leaky' catchment?

Line 613: repetition of the underestimated GPDs.

Figure 6: please keep the colours for each model matching to those in figure 5.

Figure 14: What's the difference between a and c? Does figure e include the glacier cover, or is it only snow cover? If it's only snow cover, I don't see how in figure (e) MODIS and SPHY look so similar for august, when the snow cover (green only) look quite different in b and d.

---

## Author Comment (AC2) · 21 Oct 2020

Dear Dr. Muhammad Ashraf,

Thank you very much for the positive assessment of the paper and suggestive comments. Your suggestions are very useful to improve the presentation of the paper.

Thank you for highlighting the effect of topography on precipitation distribution in the region. The correction factors for the adjusted observed precipitation are based on the elevational distribution and hydrological balance of the region (Dahri et al. 2018). The

effect of elevation on the precipitation has been well demonstrated in previous studies (Immerzeel et al. 2015, Shafeeque et al. 2019). We will incorporate the important discussions to highlight further the effect of topography on precipitation and its role in classifying the hydrological behavior of the basin.

In the revised version of the paper, we will enhance the discussion section and better explain the corrections of each dataset. We will further explore the directions for future research in this regard.

Moreover, we will deepen the discussion to explore the suitability of different gridded precipitation datasets for the sub-catchments of UIB (Hunza, Shigar, Shyok, Astore, Gilgit, Shingo, Zanskar, and Upper Indus) in the Karakorum, Hindukush, and Himalaya for the selection of suitable one based on primary influential precipitation system (westerlies and monsoon).

References:

Dahri, Z.H., Moors, E., Ludwig, F., Ahmad, S., Khan, A., Ali, I. and Kabat, P. (2018) Adjustment of measurement errors to reconcile precipitation distribution in the high-altitude Indus basin. International Journal of Climatology 38(10), 3842-3860.

Immerzeel, W.W., Wanders, N., Lutz, A.F., Shea, J.M. and Bierkens, M.F.P. (2015) Reconciling high-altitude precipitation in the upper Indus basin with glacier mass balances and runoff. Hydrology and Earth System Sciences 19(11), 4673-4687.

Shafeeque, M., Luo, Y., Wang, X.L. and Sun, L. (2019) Revealing Vertical Distribution of Precipitation in the Glacierized Upper Indus Basin Based on Multiple Datasets. Journal of Hydrometeorology 20(12), 2291-2314.

---

## Author Comment (AC3) · 21 Oct 2020

**1 General Comments**

Thank you very much for the positive assessment of our manuscript. The comments and suggestions are constructive to improve the presentation of the article. We will consider and incorporate these suggestions into the revised version of the manuscript. Following is the early point to point response to the comments and suggestions:

**2 Major comment**

[Figure]

The physical diagnosis aims to determine whether a basin can be considered plausibly realistic using a water-energy balance.

The approach is simple and easy to apply with observed physical data (discharge, precipitation, evapotranspiration). These data are usually frequently available because, in every catchment, these data are measured regularly (Andréassian and Perrin, 2012). The potential evapotranspiration is calculated using temperature and other physical data, representing the surface energy balance.

The Budyko theory-based equations describe the relationship primarily between the natural precipitation and evapotranspiration, while neglecting the changes in water storage in a watershed. This is generally a very good approximation for the non-glacierized watershed over the annual or inter-annual scale. However, for the glacierized watershed, it may fail. It is right the weakness of this approximation for the non-glacierized watershed that can be used as a tool for identifying if the water balance of a watershed is involved by the glacier melt as an additional input.

The Truc-Budyko equation assists in identifying the physical realism of the catchment under a particular precipitation dataset. In glacierized catchments, meltwater also plays a vital role in overall water balance. Baseflow contributions to the total runoff (Alley et al., 2002; Andermann et al., 2012; Savoskul and Smakhtin, 2013) compensate for the loss due to the groundwater percolation balance; therefore, the only imbalance in water balance would be due to meltwater (Immerzeel et al., 2015). Hence, the 'gaining' catchments highlight the meltwater contributions to the total runoff, whereas, the 'leaky' catchments can highlight the positive mass balance or glacier advance. The water balance equation, which has a mass balance component (Eq. 8), helps the actual under- or over-estimation in the selected/included GPDs, quantitatively.

Therefore, we are hesitated to remove this component from the Three-stage diagnosis approach. We hope it can be kept. Yet, we need to make further efforts in the composition of this section. We will present this part more clearly and concisely. Meanwhile,

we will make our efforts to concise the whole manuscript.

3 Minor Suggestions

Suggestion 1:

I should note that I was not able to see any of the figure or table references in the manuscript, which may have led to some misunderstandings on my part.

Response:

We are extremely sorry for the inconvenience. The in-text references of figures and tables were misplaced while moving figures and tables to the bottom of the text. We will update the references in the revised version.

Suggestion 2:

Given that not all readers will necessarily be familiar with the statistical plots presented here, it would be useful to have more information in the figure captions, and to define the acronyms used in the figures. E.g. for figure 8 "the boxes represent …..." e.g. for figure 10 "each dot represents …. (an average/total for one year of data?)".

Response:

Thank you for the suggestions. The figure captions will be updated in the revised version, as suggested.

Suggestion 3:

Throughout the manuscript, the authors should check that acronyms and abbreviations are defined where they first appear, and consider repeating these, or providing a nomenclature. For example on line 95, Q, P, ET_p all need defining, on line 210 ETCCDI needs defining.

Response:

Thank you. We will improve the statement of acronyms and abbreviations.

[Figure]

Suggestion 4:

Line 174: Was this adjusted using the same wind data as used in Dahri 2018? I think given the importance of this adjustment, an extra line explaining it would be useful.

Response:

The precipitation was adjusted based on the procedure and correction factors in Dahri et al. (2018). We will add this explanation of the procedure in the revised manuscript.

Suggestion 5:

Line 300: Please add references to the sentence 'The observed mass balance data were extracted from the literature'. Response:

Thank you for the suggestion. The references are also provided in the Table 2. We will add references in the main text too.

Suggestion 6:

Line 305: Where is the data from Besham Qila held or is there a reference for this data?

Response:

The discharge data at Besham Qila was collected from the Water and Power Development Authority (WAPDA), Pakistan. The reference for the data is provided in Table 2. The data from WAPDA are also acknowledged in the acknowledgment section.

Suggestion 7:

Line 323/Figure 5: Could you plot the points of the stations onto the map in figure 5 so it's easier to compare the observations and colour maps? More importantly, please switch the colour scheme for either figure 3 or 5 so that they are the same, with the same scale (a divergent colour scheme for both would make it clearer which areas are high precipitation and which are low precipitation). It's tricky to compare them when

blue is dry regions in one plot and wet regions in the other.

Response:

Thank you very much for your constructive suggestions. We will update the Figure 3 and Figure 5 following your suggestions. In revised figures blue will represent the wet regions.

Suggestion 8:

Line 326: While it's clear what you mean here, I think technically this should be 'the GDPs did not show statistically significant trends', as you cannot generally use statistical tests to prove a lack of trend.

Response:

Thank you for your suggestion. We will update the sentence in the revised version, as suggested.

Suggestion 9:

Line 331-333: Are these discussions about bias coming from the Taylor diagram in figure 6? It might be better to talk about RMSE, as that's what you have shown here in figure 6.

Response:

Thank you very much for your suggestion. We will update the explanation of Figure 6 in the text focusing on RMSE.

Suggestion 10:

Line 333: word missing '….as the better in UIB…' -> '…as the better model in the UIB…'

Response:

Thank you for the correction.

Suggestion 11:

Line 345: It would be good to emphasize that this under/overestimation is particularly true during the winter, as it's interesting that these datasets appear to represent the summer monsoon much more effectively than the winter westerlies.

Response:

Thank you for the suggestion. We will update the explanation in the revised manuscript, as suggested.

Suggestion 12:

Lines 349-355: You could consider cutting figure 7 b and the accompanying text, as I think this is all shown in figure 7 a and that discussion. If figure 7 b is kept, could the seasons be put in order? I'd recommend Winter, spring, summer, autumn as this will make it easier to see the winter and spring precipitation together.

Response:

Thank you. We will update the figure 7b in the revised version by re-ordering the seasons as recommended.

Suggestion 13:

Lines 357-368: It's not quite clear how these numbers relate to figure 8, or how the values in figure 8 and the numbers in the text are calculated. Are the mean and standard deviations given in the text taken from each year? I.e. the maximum CDD taken from a year, and then averaged over all the years? Given figure 8 shows the median and 25th/75th percentiles, it might be more useful to discuss those? (although presumably the red dots are the means in each case).

Response:

Thank you for highlighting it. Yes, the maximum CDD was taken from a year, and then averaged over all the years. Meanwhile, we will improve the presentation by discussing the boxplots' statistics (the median and 25th/75th percentiles).

Suggestion 14:

Figure 9/lines 372-382: is the runoff value the same for each of the models? Is this a measured value? Please state in the text and the figure caption.

Response:

Yes, the observed runoff values are the same for each model. We will add an explanatory statement here for clarity.

Suggestion 15:

Lines 383-391: This section discusses correlation between runoff and precipitation. However given there is no significant correlation between the observed precipitation and runoff, except in the Karakoram, it seems that there may be other factors that need to be taken into consideration, and therefore that correlation between these two variables probably should not be used to judge the datasets?

Response:

Thank you for highlighting an important point. The correlation between precipitation and runoff may not represent the validity of a precipitation dataset, and thus, should not be used alone to judge the datasets. This motivates to use 'physical diagnosis' for determining the physical realism of a catchment under different precipitation datasets. We also used a water balance equation (Eq. 8), including the mass balance component, to assess a precipitation datasets' validity.

Suggestion 16:

Line 456: Are these six SPHY model runs identical, except for the precipitation dataset used? Or are there some differences, apart from the precipitation datasets?

[Figure]

Response:

Yes, these six SPHY models are almost identical to the base calibrated model. There was a little tuning of DDFs for ice and snowmelt to keep the average mass balance closer for all the precipitation datasets. This was done to compare the rationality of the mass balance and streamflow outputs.

Suggestion 16:

Line 494-497: I don't quite understand this.

Response:

The rationality of runoff and mass balances simultaneously means that the simulated runoff and mass balance represent the actual situation (observed values) in the catchment at the same time. As the datasets are under- or over-estimated; therefore, one of these two variables (runoff or mass balance) will be under- or over-estimated accordingly (i.e., in simulated).

Suggestion 17:

Line 510: Could you add some references here?

Response:

Several researchers have evaluated the precipitation datasets using simple statistical methods (Ali et al., 2017; Anjum et al., 2018; Blacutt et al., 2015; Henn et al., 2018; Hu and Chen, 2018; Hu et al., 2016; Hussain et al., 2017; Romilly and Gebremichael, 2011). However, when the observed data do not represent a catchment's entirety, then such simple statistical methods may not provide the true assessment of precipitation datasets. The data is not sufficient (a small number of stations) and not sufficient quality (uneven distribution of stations – mostly at lower elevations) (Winiger et al., 2005). Several other researchers also pointed out these limitations in previous studies (Dahri et al., 2018; Immerzeel et al., 2015).

Suggestion 18:

Line 589-594: you suggest undercatch in the observations here as a reason for the unbalanced water balance, but haven't you already corrected for this?

Response:

In several gridded datasets, observed precipitation (uncorrected) is used, for example, APHRODITE (Yatagai et al., 2012). Therefore, it can be argued that the undercatched precipitation (in GPDs) might be a reason for an unbalanced water balance. The adjusted average precipitation in UIB represents a plausible water balance.

Suggestion 19:

Lines 595-598: is increasing glacier mass balance also a reason for a 'leaky' catchment?

Response:

Yes, it is. A part of precipitation is lost to keep the mass balance increasing (positive mass balance/glacier advance). That missing (frozen) water is not contributing to the total runoff at the moment. So, the energy balance goes beyond the limits and makes the catchment 'leaky'. We will incorporate the suggestion. Thank you.

Suggestion 20:

Line 613: repetition of the underestimated GPDs.

Response:

Thank you for highlighting. We will fix it in the revised version.

Suggestion 21:

Figure 6: please keep the colours for each model matching to those in figure 5.

Response:

Thank you for the suggestion. We will fix this in the revised Figures.

Suggestion 22:

Figure 14: What's the difference between a and c? Does figure e include the glacier cover, or is it only snow cover? If it's only snow cover, I don't see how in figure (e) MODIS and SPHY look so similar for august, when the snow cover (green only) look quite different in b and d.

Response

In Figure 14, 'a' is the MODIS maximum snow cover in March, and 'b' is SPHY maximum snow cover in March. The snow cover is slightly different in the two maps. Most of the differences are spotted at low elevations over the southeast and southwest parts. The glacier cover is also included in figure part 'e'.

We are grateful for your precious time, positive assessment, and constructive suggestions. These will improve the presentation of the research article.

Sincerely,

Yi Luo and Muhammad Shafeeque

4 References

Ali, A. et al., 2017. Evaluation and Comparison of TRMM Multi-Satellite Precipitation Products With Reference to Rain Gauge Observations in Hunza River Basin, Karakoram Range, Northern Pakistan. Sustainability, 9(11): 1954-1972. DOI:10.3390/su9111954

Alley, W.M., Healy, R.W., LaBaugh, J.W., Reilly, T.E., 2002. Flow and storage in groundwater systems. Science, 296(5575): 1985-90. DOI:10.1126/science.1067123

Andermann, C. et al., 2012. Impact of transient groundwater storage on the discharge of Himalayan rivers. Nature Geoscience, 5(2): 127-132. DOI:10.1038/ngeo1356

Andréassian, V., Perrin, C., 2012. On the ambiguous interpretation of the Turc-Budyko nondimensional graph. Water Resources Research, 48(10): W10601. DOI:10.1029/2012wr012532

Anjum, M.N. et al., 2018. Performance evaluation of latest integrated multi-satellite retrievals for Global Precipitation Measurement (IMERG) over the northern highlands of Pakistan. Atmospheric Research, 205(October 2017): 134-146. DOI:10.1016/j.atmosres.2018.02.010

Blacutt, L.A., Herdies, D.L., de Gonçalves, L.G.G., Vila, D.A., Andrade, M., 2015. Precipitation comparison for the CFSR, MERRA, TRMM3B42 and Combined Scheme datasets in Bolivia. Atmospheric Research, 163: 117-131. DOI:10.1016/j.atmosres.2015.02.002

Dahri, Z.H. et al., 2018. Adjustment of measurement errors to reconcile precipitation distribution in the high-altitude Indus basin. International Journal of Climatology, 38(10): 3842-3860. DOI:10.1002/joc.5539

Henn, B., Newman, A.J., Livneh, B., Daly, C., Lundquist, J.D., 2018. An assessment of differences in gridded precipitation datasets in complex terrain. Journal of Hydrology, 556: 1205-1219. DOI:10.1016/j.jhydrol.2017.03.008

Hu, Z., Chen, D., 2018. Evaluation of three global gridded precipitation data sets in central Asia based on rain gauge observations. (February): 1-19. DOI:10.1002/joc.5510

Hu, Z., Hu, Q., Zhang, C., Chen, X., Li, Q., 2016. Evaluation of reanalysis, spatially interpolated and satellite remotely sensed precipitation data sets in central Asia. Journal of Geophysical Research: Atmospheres, 121(10): 5648-5663. DOI:10.1002/2016jd024781

Hussain, S. et al., 2017. Evaluation of gridded precipitation data in the Hindu Kush–Karakoram–Himalaya mountainous area. Hydrological Sciences Journal, 62(14): 2393-2405. DOI:10.1080/02626667.2017.1384548

Immerzeel, W.W., Wanders, N., Lutz, A.F., Shea, J.M., Bierkens, M.F.P., 2015. Reconciling high-altitude precipitation in the upper Indus basin with glacier mass balances and runoff. Hydrology and Earth System Sciences, 19(11): 4673-4687. DOI:10.5194/hess-19-4673-2015

Romilly, T.G., Gebremichael, M., 2011. Evaluation of satellite rainfall estimates over Ethiopian river basins. Hydrology and Earth System Sciences, 15(5): 1505-1514. DOI:10.5194/hess-15-1505-2011

Savoskul, O.S., Smakhtin, V., 2013. Glacier systems and seasonal snow cover in six major Asian river basins: hydrological role under changing climate, 150. IWMI.

Winiger, M., Gumpert, M., Yamout, H., 2005. Karakorum-Hindukush-western Himalaya: assessing high-altitude water resources. Hydrological Processes, 19(12): 2329-2338. DOI:10.1002/hyp.5887

Yatagai, A. et al., 2012. APHRODITE: Constructing a Long-Term Daily Gridded Precipitation Dataset for Asia Based on a Dense Network of Rain Gauges. B Am Meteorol Soc, 93(9): 1401-1415. DOI:10.1175/BAMS-D-11-00122.1